



# Winter atmospheric nutrients and pollutants deposition on West Sayan mountain lakes (Siberia)

Daniel Diaz-de-Quijano[1]; Aleksander Vladimirovich Ageev[1]; Elena Anatolevna Ivanova[1]; Olesia Valerevna Anishchenko[1,2]

[1]Siberian Federal University, 79, Svobondyi prospekt, Krasnoyarsk, 660041, Krasnoyarskii Krai, Russian Federation.

[2]Institute of Biophysics, Siberian Branch, Russian Academy of Sciences, 50/50, Akademgorodok, Krasnoyarsk, 660036, Krasnoyarskii Krai, Russian Federation.

*Correspondence to*: Diaz-de-Quijano, D. (ddiasdekikhanobarbero@sfu-kras.ru, daniquijano@gmail.com)

**Abstract.** The world map of anthropogenic atmospheric nitrogen deposition and its effects on natural ecosystems is not
described with equal precision everywhere. In this paper, we report atmospheric nutrient, sulphate and spheroidal carbonaceous particles (SCPs) deposition rates, based on snowpack analyses, of a formerly unexplored Siberian mountain region. Then, we discuss their potential effects on lake phytoplankton biomass limitation.

We estimate that the nutrient depositions observed in the late season snowpack ($40\pm16$ mg $NO_3$-N$\times m^{-2}$ and $0.58\pm0.13$ mg TP-P$\cdot m^{-2}$) would correspond to yearly depositions lower than $119\pm71$ mg $NO_3$-N$\cdot m^{-2}\cdot y^{-1}$ and higher than $1.71\pm0.91$ mg TP-P$\cdot m^{-2}\cdot y^{-1}$. These yearly deposition estimates would approximately fit the predictions of global deposition models and correspond to the very low nutrient deposition range although they are still higher than world background values.

In spite of the fact that such low atmospheric nitrogen deposition rate would be enough to induce nitrogen limitation in unproductive mountain lakes, the extremely low phosphorus deposition would have made the bioavailable N:P deposition ratio to be frankly high. In the end, lake phytoplankton appeared to be hanging on the fence between phosphorus and nitrogen
limitation, with a trend towards nitrogen limitation. We conclude that slight imbalances in the nutrient deposition might have important effects on the ecology of these lakes under the expected scenario of climate warming, increased winter precipitation, enhanced forest fires and shifts in anthropogenic nitrogen emissions.

## 1 Introduction

Worldwide nitrogen cycle perturbation is the second most important global environmental concern, just after massive extinction of species and even more important than global warming (Rockström et al., 2009; Steffen et al., 2015). The anthropogenic mobilization of formerly inaccessible nitrogen compartments has more than doubled natural nitrogenase-



mediated inputs of reactive nitrogen forms into the global nitrogen cycle (Vitousek et al., 1997). Massive fossil fuel combustion since the industrial revolution, chemical fixation of atmospheric diatomic nitrogen to produce fertilizers since the Second

World War and the wide extension of leguminous crops are the most important human sources of nitrogen cycle perturbation (Vitousek et al., 1997). A substantive part of this anthropogenic reactive nitrogen is then spread, air-transported and deposited all over the world with a diverse impact on different ecosystems.

The effects of atmospheric nitrogen deposition on primary production have been documented not only in the paradigmatically nitrogen-limited terrestrial ecosystems (Bobbink et al., 2010; DeForest et al., 2004; Güsewell, 2004; LeBauer and Treseder,

2008), but also in the paradigmatically phosphorus-limited lakes (Bergström et al., 2005). A series of studies all over Sweden and abroad showed atmospheric nitrogen deposition to have turned unproductive lake phytoplankton from natural nitrogen to induced phosphorus limitation (Bergström and Jansson, 2006; Bergström et al., 2005; Elser et al., 2009) when temperature was not a limiting factor (Bergström et al., 2013). Of course, these changes do not only concern primary production limitation, but also primary producer species composition, cascade effects over the food web, secondary production, species interactions,

etc. Likewise, these studies showed that it was reasonable to study the relationship between atmospheric nutrient deposition and lake phytoplankton growth limitation independently from biogeochemical processes occurring at the levels of the watershed, runoff and river transport, lake sediments, etc.

Nevertheless, world ecology "does not occur on a needle tip", as Ramon Margalef used to say (Bascompte and Solé, 2005; Margalef, 1986). There is a particular and dynamic geography of reactive nitrogen sources, an atmospheric conveyor belt with

a conspicuous structure, an evolving climate with patchy temperature and precipitation changes, and a multiplicity of lake districts with distinct individual lakes in them. If it is true that climatic and atmospheric nutrient deposition models have helped a lot to describe this geography, the latter ones lack empirical measurements for some regions of the world, which might undermine their regional spatial reliability in comparison to climate models (Lamarque et al., 2013; Mahowald et al., 2008). Moreover, not all lake districts of the world have been studied with the same intensity, so certain processes might be overlooked

and the limnological paradigms might be site-biased (Marcé et al., 2015). In this study, we analysed the snowpack in the West Sayan mountains (south central Siberia) in order to gauge atmospheric nitrogen, phosphorus, sulphate and spheroidal carbonaceous particles (SCPs) deposition rates. As far as we know, no such measurements had been pursued in this site before, so they might be useful to contrast and inform world deposition models. Besides, we have also assessed lake phytoplankton nutrient limitation regime and discussed the potential influence of nutrient deposition on it.

According to published global models (IPCC, 2013; Lamarque et al., 2013), the West Sayan mountains, in south central Siberia, correspond to a cold but notably warmed and relatively low atmospheric nitrogen deposition area. Our aim was to corroborate it because in case it was confirmed, it would be an adequate site to study the effects of global warming on ecosystems with a minimal interference of atmospheric nitrogen deposition. In other words, identifying and studying such areas could help disentangle warming and nitrogen fertilization as drivers of ecological change. It could also contribute to

assess the worthiness to implement global nitrogen cycle policies, besides climate ones.



## 2 Methods

### 2.1 Study site and sampling

West Sayan mountain range is located in south central Siberia. It has a central position in the Altay-Sayan mountain system, in between the Altay mountains (to the west) and East Sayan mountains (to the east), which are constituents of the Sayan-Baikal mobile fold belt south the Siberian craton (Logatchev, 1993). West Sayan orogeny occurred in the ancient Paleozoic, by folding Paleozoic and Precambrian deposits, during the Baikal tectogenesis and in the Cenozoic era (namely during the Pliocene-Pleistocene Epochs) (Chernov et al., 1988). With a north-west orientation and heights from 400 to 2700 m.a.s.l., West Sayan mountains combine old eroded with typical glacial reliefs, carved during the Pleistocene glaciation in the highest ridges. The source of Yenisei river, the first Siberian river in terms of discharge, is located in West Sayan mountains and its headwater tributaries are also Sayanic.

The present study was performed in the Ergaki Natural Park, in the West Sayan mountains. With an altitude range from 700 to 2466 m.a.s.l., this park is well known for the glacial landscapes of both Ergaki and Aradan ridges embedded in a boreal mountainous taiga matrix, that extends far to the north. The landscape is spattered with monumental and pictoric granite-syenite rocks, and the general geology is rich in granitoids (Voskresenskii, 1962). South from the park, sub-boreal larch taigas and central Asian steppes develop. The closest gardens and agricultural fields are located downhill more than 35 Km north from the northernmost sampling point and constitute a modest patch within the taiga matrix. Climate in the Ergaki Natrual Park is characterized by high precipitation (1243 mm) and extreme temperatures, ranging from -36.8 to +33.3°C (fig. 1 a). From a geobotanical point of view the park is located in the holarctic kingdom, circumboreal province and Altay-Sayan province (Takhtadzhyan, 1978). Lowland deciduous birch (*Betula pendula* Roth.) and aspen (*Populus tremula* L.) forests are succeeded by pine (*Pinus sylvestris* L. 1753) and larch (*Larix sibirca* Ledeb. 1833) light taiga which extends from the lowlands to an altitude of about 1300-1400 m, although the red pine is also present at higher altitudes. The 1300-1700m range is occupied by a darker taiga composed of firs (*Abies sibirica* Ledeb. 1833), Siberian pine (*Pinus sibirica* Du Taur, 1803) and to a lesser extent spruce (*Picea obovata* Ledeb. 1833). At altitudes higher than 1600-1700 there is a subalpine forest composed of Siberian pine patches with an undergrowth of dwarf birch (*Betula nana* L. subsp. *rotundifolia* (Spach.) Malyschev, 1965) and *Rhododendron adamsii* Rehd. 1921 (Shaulo, 2006). Montane and alpine meadows, peatlands, mountain tundra and snowdrift vegetation are also present in the study area. An exhaustive floristic description is available at (Stepanov, 2016), where our study area corresponds to the L3 district. Interesting facts are remarked in this monography, such as the presence of southern fabaceae species originary from arid central Asia.

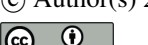



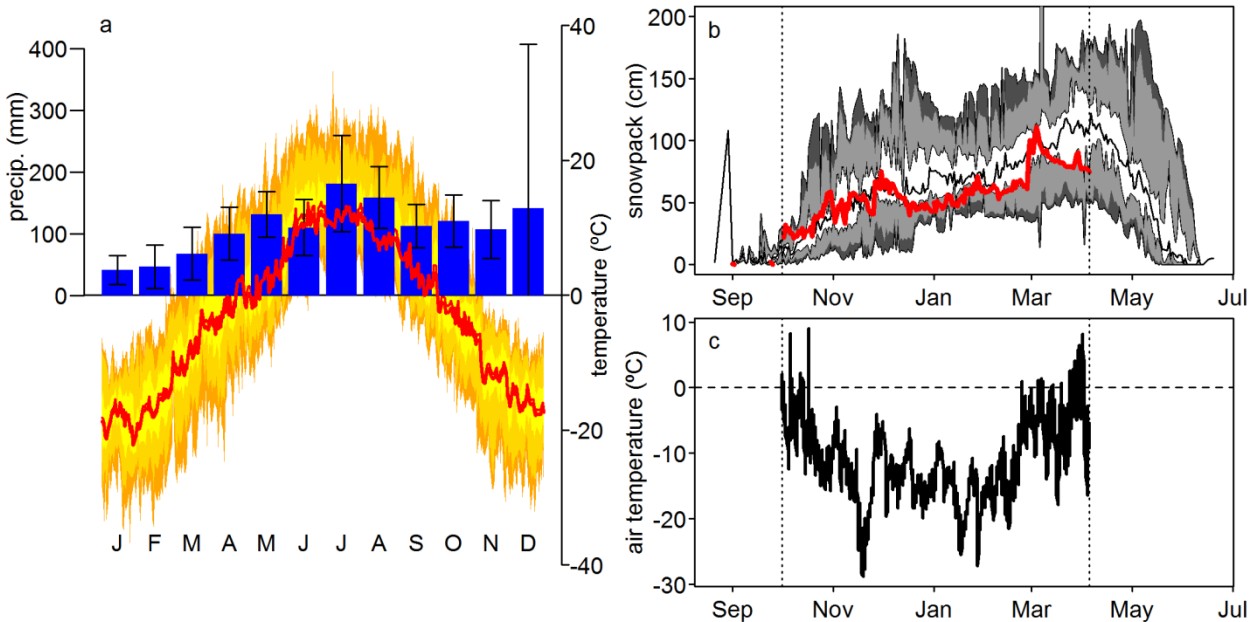

**Figure 1. Climatic characteristics at Olenya Rechka meteorological station. a: Climograph (1 February 2005 - 30 April 2019). The median yearly precipitation was 1242.975 mm, with a winter (2nd October to 5th April) and summer precipitations of 464.95 mm and 778.025 mm, respectively. Temperature: median (red), interquartile range (yellow band), 5th to 25th percentiles and 75th to 95th percentiles (lower and upper golden bands), below the 5th percentile and above the 95th percentile (lower and upper orange bands). b: Snowpack thickness (2005-2017): median (black), interquartile range (white band), 5th to 25th percentiles and 75th to 95th**
**percentiles (lower and upper light gray bands), below the 5th percentile and above the 95th percentile (lower and upper dark gray bands). The red line corresponds to the 2016-2017 snowpack thickness record until the snow sampling date. c: Air temperatures measured all the 3 hours during the time period when analysed snowpacks were laying on their respective locations (2016-17).**

Snowpack cores were sampled at three sites of the Ergaki Natural Park: next to lake Tsirkovoe (Цирковое), next to lake Oiskoe (Ойское) and on a forest glade close to Tushkan stream (Тушкан) (fig. 2, table 1). Snow sampling was conducted the 5th of

April 2017, integrating a snowfall period of 6 months and 5 days according to precipitation data recorded in the closeby Olenya Rechka metereological station (http://rp5.ru). The 1st and 24th September 2016 snowfalls thinner than 0.5 cm and 2 cm respectively were registered but they melted the following day. The first important snowfall occurred the 1st October 2016 evening and left a 22 cm pack that was not significantly reduced anymore until the sampling day (fig. 1 b). Air temperatures recorded in the mentioned meteorological station during this time window were mostly below zero, with positive temperatures

only for some hours around midday during the first and last weeks (fig. 1 c).



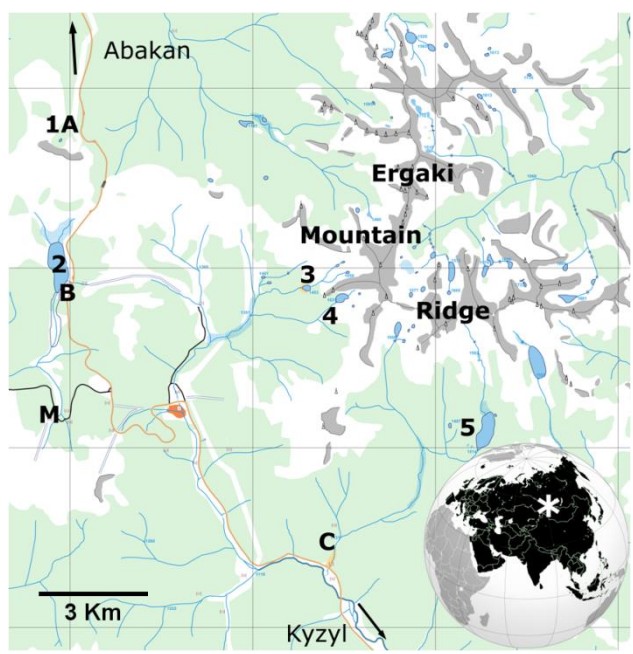

**Figure 2. Distribution of sampling points in the Ergaki Natural Park. Snow cores: Tsirkovoe (A), Oiskoe (B) and Tushkan (C); Lakes: Tsirkovoe (1), Oiskoe (2), Raduzhnoe (3), Karovoe (4), Svetloe (5); Olenya Rechka metereological station (M). Ergaki location in Eurasia. Mountain ridge (grey), open spaces (white), forest and bushes (green), three lane federal road (yellow). Source:**
**http://www.shandl.narod.ru/map.htm and Wikipedia CC BY-SA 3.0**

| | latitude | longitude | Altitude (m) | Distance in m to local perturbations | | |
| --- | --- | --- | --- | --- | --- | --- |
| | | | | road | cottage | inflow from houses |
| **Tsirkovoe** | 52°52'28.3"N | 93°14'53.1"E | 1428 | 466 | - | - |
| **Tushkan** | 52°46'16.9"N | 93°21'17.0"E | 1125 | 725 | 471 | - |
| **Oiskoe** | 52°50'28.3"N | 93°14'46.0"E | 1418 | 251 | 229 | 288 |
| **Raduzhnoe** | 52°50'08.4"N | 93°20'44.5"E | 1462 | 4600 | 3000 | - |
| **Karovoe** | 52°49'57.4"N | 93°21'41.6"E | 1632 | 5265 | 4000 | - |
| **Svetloe** | 52°48'02.2"N | 93°25'05.4"E | 1511 | 5647 | 5470 | - |

**Table 1 Sampling sites and distance to local perturbations. Temporary summer camps are present on Svetloe lake shore and used to be on Raduzhnoe's.**

The snow core sampling was conducted following a modified version of the MOLAR project protocols for atmospheric deposition assessment (Mosello et al., 1997). Sampling areas were chosen on a map to be accessible but as far as possible from
local sources of air pollution. Definitive locations were also chosen to represent average snowpack thicknesses by checking it across the sampling areas using a snow probe. Hence, wind and orography secondary modifications of the snowpack were minimised. An aluminium tube and piston (1m x 2.5 cm inner diameter), plastic shovel, plastic containers and rubber gloves

were soaked in ~4% HCl and MQ water rinsed before being used to pick up snow cores. A protective mask and synthetic clothes were worn during sampling. The snow was stored in the plastic containers and kept at -20°C until further analyses.

Two cores divided in three segments (0-40, 40-80 and 80-115 cm) were sampled at Tsirkovoe, whereas three cores divided in two segments (0-60 and 60-115 cm) were sampled at Oiskoe and Tushkan.

Lake water was sampled at different depths and consequently analysed for chlorophyll and nutrient content in early September 2015 (Tsirkovoe, Oiskoe and Raduzhnoe; Радужное) and in late August 2017 (Tsirkovoe, Oiskoe and Karovoe; Каровое). Data from a previously published study integrating June and August samplings 2011-12 (Oiskoe, Raduzhnoe, Karovoe and

Svetloe; Светлое) was also used (Anishchenko et al., 2015). Water samples were sieved *in situ* to remove zooplankton, transported to the field laboratory at 4-10°C in the dark, filtered for chlorophyll analyses and frozen at -20°C for further chemical analyses.

## 2.2 Chemical analyses

Water and snow samples were fully thawn and stirred before analyses. Snow water equivalent (SWE) was calculated by

multiplying the snowpack depth and the ratio of melted water volume to sampled snow volume. Ammonium was determined by nesslerization in samples filtered through 0.45 µm membrane filters "Porafil" (Macherey-Nagel, Germany) with single-use plastic syringes. Detection limit for ammonium was 0.014 mg·l$^{-1}$. The other dissolved chemical species ($NO_2^-$, $NO_3^-$, soluble reactive phosphorus –SRP– and $SO_4^{2-}$) were analysed on gently vacuum filtered aliquots also using the same mentioned filters. Nitrate was reduced to $NO_2^-$ by the cadmium reduction method. Nitrite was determined by the colorimetric method after

reacting with sulphanilamide and α-naphthylamine. Detection limits were 0.0006 mg·l$^{-1}$ and 0.005 mg·l$^{-1}$., for $NO_2^-$ and for $NO_3^-$, respectively. Lake water dissolved inorganic nitrogen (DIN) was calculated as the sum of nitrate, nitrite and ammonium. Soluble reactive phosphorus was assessed using the ascorbic acid and ammonium molybdate method. Total phosphorus was measured the same way after persulfate digestion of unfiltered samples. All these analyses were made according to the Russian National Standards (Gladyshev et al., 2015; Tolomeev et al., 2014), which generally coincide to those from APHA (APHA,

1989). As for $SO_4^-$ analysis, snow samples were concentrated by heating, HClO$_4$ and HNO$_3$ mixture was added and evaporated, then an ion-exchange column was used to remove interferences of cations. Samples were titrated with BaCl$_2$ solution in the presence of nitrochromazo until blue color appearance (Kalacheva et al., 2002). Finally, total nitrogen (TN) was digested from total snow samples using persulfate and boric acids and subsequently transformed into $NO_3^-$ (Grasshoff et al., 1983). The natural light absorption of this nitrate at 210 nm was determined using a Spekol 1300 photometer (Analytik Jena, Germany)

and corrected for organic matter interference by subtracting absorption at 275 nm (Slanina et al., 1976).

## 2.3 Chlorophyll and SCPs analyses

A known fraction of melted snow samples was filtered through GF/C filters to collect SCPs (Mosello et al., 1997). Nitric, hydrofluoric and chlorhydric acids were used to remove organic, siliceous and carbonate material, respectively (Rose, 1994;



Yang et al., 2001). Determinate fractions of the samples were mounted on NAPHRAX and counted at 400X under an Axiostar

plus microscope (Zeiss). Negative controls and a sediment reference standard were likewise processed to correct final counts

for any experimental bias (Rose, 2008).

Phytoplankton chlorophyll was assessed according to the UNESCO standard protocols (VA, 1997). Samples had been filtered

in the field laboratory through $BaSO_4$-covered 0.45 µm membrane filters "Porafil" (Macherey-Nagel, Germany), folded

inwards and frozen. They were then let thaw, dried in the dark, and scraped along with $BaSO_4$ into centrifuge tubes. Pigments

were extracted in 100% acetone for 9h in the dark at +4°C. After filtration through 0.2 µm polycarbonate filters, MQ water

was added to get pigments dissolved in a 90% acetone solution, f.c. Photometric measurements were used to calculate

chlorophyll concentrations (Jeffrey and Humphrey, 1975).

## 2.4 Air mass retrotrajectory analysis and statistics

The retrotrajectories of air masses flowing on the three snow sampling sites were obtained using the Hybrid Single-Particle

Lagrangian Integrated Trajectories (HYSPLIT) model for archive trajectories (Rolph et al., 2017; Stein et al., 2015) of the

National Oceanic and Atmospheric Administration Air Resources Laboratory (NOAA, USA). A total of 187 daily

retrotrajectories embracing the sampled period of atmospheric deposition were reconstructed as the snowpack bulk deposition

airshed. Each trajectory started three days back in the past. It recorded a per hour latitude, longitude and altitude coordinates

and ended up at the snow sampling coordinates, at 0 m above model ground level at 24h of consecutive days. All the analyses

were performed within the R environment (R Development Core Team, 2017). Total retrotrajectory length and average wind

speed the hour before getting to the sampling point were calculated using the Vincenty (ellipsoid) distance method within the

geosphere package (Hijmans, 2017). The openair package was used to determine wind direction and to draw wind roses

(Carslaw and Ropkins, 2012). The number of per hour coordinates at 0 m above model ground level was calculated to

characterise the direct interaction of each air mass with the Earth crust. Retrotrajectories were mapped using the ggmap package

(Kahle and Wickham, 2013). One-way ANOVA comparing sites and Pearson correlation analyses of chemical and wind

variables in the seven analysed snow core sections were performed using built-in functions of the R statistical environment.

## 3 Results and discussion

### 3.1 Potential fragmentation of nutrients by snow melting

The three sampled snow cores were 115 cm deep, but had different SWE: 25±1 cm in Tsirkovoe, 27±1 cm in Oiskoe and

12±0.3 cm in Tushkan. First of all, the snowpack temperature profile was measured to determine if snow melting could have

occurred before sampling. Major snow thawing can be discarded in any of the 3 sampling sites because snow temperature was

not around 0°C but always lower. Nevertheless, the deepest snowpack layers fall within the range between -2 and 0°C: Oiskoe

at 110 cm deep, Tsirkovoe from 90 to 110 cm, and namely Tushkan from 60 to 110 cm deep. This indicates that snow melt

was either about to occur or could have even started in these particular layers, triggering a sequential elution of solutes (Mosello





et al., 1997). In that hypothetical case, snowpack-based atmospheric deposition estimates would be biased. In order to discard such a case, solute concentrations in the upper and colder snow layers were compared to those in the deeper and warmer ones (fig. 3, table S1).





**Figure 3 Chemical composition of upper (dark grey) and lower (light grey) layers of the 2016-17 snowpack in Ergaki mountains. All values are in µg·l⁻¹ except SCPs (counts·l⁻¹). Bars represent mean values and whiskers, standard deviation. The two upper layers were averaged in Tsirkovoe, where the snow core was divided into three layers. Column pairs with "a" and "b" letters are significantly different (one-way ANOVA, p-v<0.05 in the case of nitrate; t-tests in the other cases; n=3 except in Tsirkovoe, where upper layer n=4 and lower layer n=2).**

The hypothesis was that deeper and warmer layers, suspect of possible melting, would show lower solute concentration in case of important melting, preferentially in those solutes that elute firstly during snow melting. Yet, because the first centimetres of snowpack were formed much faster than the rest of the snowpack, it is conceivable that the deep layers were originally poorer in airborne chemicals and particles, which would bother the initial hypothesis. Indeed, the first third of snowpack thickness at Olenya Rechka meteorological station deposited in only 19 days (from 10/01 to 10/19), whereas it took 39 days (from 10/01 to 11/08) to attain half of its thickness at sampling date (i.e. 187 days after initial snowpack formation). It is likely that the deepest Tsirkovoe, Oiskoe and Tushkan snow core segments (80-115 cm in the first case and 60-115 cm in the others) would have formed in about 19 and 39 days, respectively. Nevertheless, if precipitation rate had had a determinant effect on the vertical distribution of solutes and particles content, the lower values in deeper layers should be expectable in all the measured variables, and it was not the case (fig. 3).

Thus, no significant differences were found between the upper and deeper layers in any of the measured variables except for nitrate, with lower values in the deep layers (ANOVA, p-value= 7.75·10⁻⁵). The other significant differences between upper and deeper snow layers (TN in Tsirkovoe, SRP in Tushkan, TP in Oiskoe and SCPs in Tushkan; t-tests) were not consistent across sampling sites (fig. 3). Besides, sulphate also had slightly lower concentrations in the deep snow layers but this difference was not statistically significant. This is especially explanatory because preferential elution of ions during snow melt occurs either in the sequence $SO_4^{2-}$>$NO_3^-$>$NH_4^+$ (Kuhn, 2001) or $SO_4^{2-}$>$NH_4^+$>$NO_3^-$ (Wang et al., 2018a), but sulphate always elutes preferentially to inorganic nitrogen species, according to the literature (Cragin et al., 1996; Kuhn, 2001; Stottlemyer and Rutkowski, 1990; Williams and Melack, 1991). In other words, higher proportions of sulphate are released during early snow melting steps as compared to nitrate or ammonium. As a result, only significant lower values of sulphate should be observable in incipient thawing snow layers whereas both sulphate and nitrate would be significantly leaked at a more advanced thawing stage. Therefore, we suggest that the only observed differences in nitrate concentrations between layers might not be due to snow melting. Even if it is true that sulphate also tends to be lower at deep warm snow layers, the fact of being non-significant allows us to discard thawing as a cause, and entails sulphate load estimates wouldn't be thaw-biased nor any of the other solutes, which should elute at a later stage. As a conclusion, snowpack-based estimates of atmospheric deposition should always be cautiously considered, but major elution of solutes due to snow melting was not detected in the present study, probably thanks to the consistently negative temperatures along almost the whole integrated time period.

### 3.2 Snow nutrients and pollutants composition

Nutrient concentrations in Ergaki snowpack (table 2, table S1) generally take intermediate positions in comparison with other snowpack studies around the world. For instance the average 191±35 µg NO₃-N·l⁻¹ in Ergaki is higher than an old record in

the Pyrenees (115±106 µg $NO_3$-N·$l^{-1}$, Catalan 1989) but lower than a bit more recent one in the same mountains (280 µg $NO_3$-N·$l^{-1}$, Felip et al. 1995). It also takes an intermediate position relative to the Alps: lower than in Tyrolean Alps (308 µg $NO_3$-

N·$l^{-1}$, Felip et al. 1995) but higher than most sampling points in the French Alps (Dambrine et al., 2018). Finally, nitrate concentration in Ergaki snow was in between that of the Bothnian Bay of the Baltic sea (480 ±130 µg $NO_3$-N·$l^{-1}$, Rahm et al. 1995) and the lake Tahoe basin in Sierra Nevada (14-138 µg $NO_3$-N·$l^{-1}$, Pearson et al. 2015). Note that, paradoxically, the former is considered a low atmospheric nitrogen deposition region (Bergström and Jansson, 2006) whereas the latter has been reckoned as an airborne nutrient enriched area (Sickman et al., 2003) where atmospheric nitrogen deposition has shifted

phytoplankton limitation from N and P colimitation to persistent P limitation (Jassby et al., 1994).

|  | | NH4-N | NO2-N | NO3-N | TN | PO4-P | TP | SO4-S | SCPs |
|---|---|---|---|---|---|---|---|---|---|
| **Average concentration in snow** | | n.d. | n.d. | 191 (34) | 483 (165) | 2.55 (2.13) | 3.33 (2.42) | 864 (106) | 805 (275) |
| **Half year deposition** | | n.d. | n.d. | 40 (16) | 97 (56) | 0.43 (0.15) | 0.58 (0.13) | 190 (91) | 159 (48) |
| **Deposition rate (~time)** | | n.d. | n.d. | 79 (47) | 191 (132) | 0.84 (0.48) | 1.13 (0.60) | 372 (236) | 312 (174) |
| **Deposition rate (~precipitation)** | | n.d. | n.d. | 119 (71) | 288 (198) | 1.26 (0.73) | 1.71 (0.91) | 560 (356) | 470 (262) |

**Table 2. Average concentrations, half year depositions and estimated yearly deposition rates, as averaged by the 3 sampled sites (µg·$l^{-1}$, mg·$m^{-2}$ and mg·$m^{-2}$·$y^{-1}$, respectively) (SCPs in counts·$l^{-1}$, $10^3$ counts·$m^{-2}$ and $10^3$ counts·$m^{-2}$·$y^{-1}$). Mean values are shown, standard deviation in parenthesis, "n.d." means non detected.**

Total nitrogen and total phosphorus in Ergaki snowpack (table 2) were higher than in the first mentioned Pyrenean study but

lower than in the Baltic: 194±135 and 1054±363 µg TN-N·$l^{-1}$, and 2.38±0.59 and 9.3±5.1 µg TP-P·$l^{-1}$, respectively. Total phosphorus concentration also was within the lowest range of that measured around lake Tahoe (3-109 µg TP-P·$l^{-1}$, Pearson et al. 2015). Nevertheless, ammonium and nitrite were undetectable in the present study but detected in most of the previous studies in the snowpack (e.g. Catalan 1989; Pearson et al. 2015). Ammonium was also detected in snow surrounding the city of Krasnoyarsk by our own lab, using the same analytical method as here (unpubl.). It is very likely that ammonium

concentrations in the present study were under the detection limit, as nitrate values were more than five-fold lower than in Krasnoyarsk city snow samples, where ammonium had been detected. Finally, nutrient bioavailability is an attribute of the Ergaki snowpack as 77% TP was in the form of phosphate and about 42% TN was nitrate.

Besides ammonium and nitrite, sulphate concentrations in Ergaki snowpack were also a little bit unusual. Sulphate was the most abundant of the measured ions. It doubled that in the Pyrenees in the late eighties (401±106 µg $SO_4$-S·$l^{-1}$, Catalan 1989)

and quadrupled that on lake Tahoe (Pearson et al., 2015). Sulphate concentration in Ergaki snowpack was only similar to the





highest values in the literature for non-urban areas, such as on the south coast of lake Superior in the eighties (828±216 µg SO$_4$-S·l$^{-1}$ in average, Stottlemyer and Rutkowski 1990). Altogether, nitrogen and phosphorus concentrations reached intermediate-low values but sulphate concentration was remarkably high in Ergaki snowpack.

### 3.3 Atmospheric deposition load

Roughly half year cumulative deposition corresponding to the snow season -187 days- is summarized in table 2 (second row; table S1). Unfortunately, snow-free season depositions were not measured in the present study and, consequently, yearly deposition rates could not be determined. Nevertheless, preliminary estimations were conducted assuming either a constant deposition rate along the year –time-weighted estimate– or a precipitation-weighted deposition rate (table 2, 3$^{rd}$ and 4$^{th}$ rows, respectively). These assumptions entail different simplifications concerning the seasonal pattern of emission, transport and

deposition of the different chemical species in this particular part of the world. The precipitation-weighted estimate should be, *a priori*, a more accurate estimate because wet deposition is known to be the main contributor to total deposition. Indeed, the accumulated precipitation registered in Olenya Rechka meteorological station during the studied snow season was 419 mm, whereas almost the double (819 mm) were registered during the following months up to complete a year. Note that the 2016-17 seasonality was a bit more prominent than the median 2005-2019 seasonal precipitation (fig. 1 a). Nevertheless, both

estimations neglect the emission seasonality, which might turn the constant deposition estimate into the most credible one in some cases.

To evaluate our different estimates, we compared them to seasonal depositions in the literature and discussed their likely seasonal emissions. As a rule of thumb, weak seasonality is observed for chemical species with low deposition loads. This was clearly the case of atmospheric phosphorus deposition. Atmospheric phosphorus depositions are particularly low in taiga

landscapes like Ergaki, where spring and summer biogenic aerosols –mainly pollen– represent the largest share of atmospheric phosphorus sources (Banks and Nighswander, 2000; Doskey and Ugoagwu, 1992; Mahowald et al., 2008; Wang et al., 2015). Additionally, our study site has a much higher precipitation during the snow-free season. The co-ocurrence of biogenic aerosols and almost two thirds of the precipitation during the snow-free season implies that even our precipitation-dependent estimates (table 2, 4$^{th}$ row) might be underestimates, as biogenic aerosols are not taken into account. In order to evaluate the magnitude of our underestimation,  similar studies in cold forest landscapes and with seasonal resolution were checked. Although we

didn't find any study with seasonal resolution and snow season atmospheric phosphorus loads as low as those measured in Ergaki, some clues were given by a bunch of sampling sites around the lake of Bays (ON, Canada) (Eimers et al., 2018), lake Simcoe (ON, Canada) (Brown et al., 2011), and a Tibetan forest (Wang et al., 2018b). Phosphorus snow period loads were about 9, 11 and 18 times larger than in Ergaki, snow-free season atmospheric phosphorus depositions, 2.4, 5 and 7.4 times higher than in the snow season, and snow-free season precipitations, 0.88, 0.91 and 9.1 times that of snow season, respectively.

In conclusion, the magnitude of snow season phosphorus load, and the seasonalities of precipitation and phosphorus deposition were positively correlated, and the snow-free to snow season phosphorus deposition factor could be guessed, in principle, for Ergaki. Nevertheless, the available data is too scarce to make any formal prediction based on a multiple nonlinear regression.



If phosphorus deposition seasonality strictly depended on the yearly phosphorus load, the seasonal factor should be much
lower than 2.4. On the other hand, if phosphorus deposition seasonality just depended on precipitation seasonality, Ergaki precipitation seasonality (1.95) would correspond to a phosphorus deposition seasonality of about 5.5. The latter factor (5.5) would definitely provide an overestimation of snow-free season phosphorus deposition, whereas it is not certain if the former one (2.4) would either over- or underestimate it. If we applied 2.4 and 5.5 factors to estimate snow-free season phosphorus deposition and added it to measured snow season deposition, in order to calculate yearly deposition rates they would be: 1.972
mg TP-P· $m^{-2}$·$y^{-2}$ and 1.462 mg PO4-P· $m^{-2}$·$y^{-2}$, and 3.77 mg TP-P· $m^{-2}$·$y^{-2}$ and 2.795 mg PO4-P· $m^{-2}$·$y^{-2}$, respectively. For all the above-mentioned reasons, it is very likely that the actual yearly load in Ergaki was safely below the latter estimates, whereas the former ones are not far above the precipitation-dependent estimates (table 2, 4[th] row) and, possibly, represent a more realistic guess.

An alternative even more inaccurate option to gauge the likelihood of these estimates would be to multiply pollen deposition
taxes in south Siberian sites similar to Ergaki by the phosphorus content of the most abundant pollen grains. Pollen grain weight and its specific total phosphorus content was averaged for different species of the genuses *Pinus*, *Abies*, *Picea*, *Larix* and *Betula* (Banks and Nighswander, 2000; Bigio and Angert, 2018; Brown and Irving, 1973; Doskey and Ugoagwu, 1992). The pollen deposition rates in the sediment of 3 south Siberian lakes were considered: lake Teletskoye in south west Siberia and lakes Arangatui and Dulikha on the central and south coast of the Baikal lake, south-eastern Siberia (Andreev et al., 2007;
Bezrukova et al., 2005). Their respective pollen contribution to yearly total phosphorus deposition would be 14.3, 0.398 and 2.6 mg TP-P· $m^{-2}$·$y^{-2}$. This inaccurate approach includes factors other than deposition, such as watershed to lake area, vegetation coverage and composition, wind regime, etc. that might be different to the ones at our study site. Nevertheless, the resulting wide range of values suggests that the abovementioned estimate of 1.972 mg TP-P· $m^{-2}$·$y^{-2}$, which implied a pollen contribution of 0.263 mg TP-P· $m^{-2}$·$y^{-2}$ above the precipitation-dependent estimate, could either be a credible value or an
underestimate.

As for atmospheric nitrate deposition, no significantly differences were observed between autumn-winter and spring-summer seasons in natural forested areas like Ergaki (Kopáček et al. 2011b; Xu et al. 2018). A simplistic conclusion would be to think that this constant atmospheric nitrate deposition was also true in Ergaki, where the measured winter nitrate deposition was five and four times lower than in the south Bohemian forest and the Chinese "background" sites, respectively. Then, our time-
weighted deposition estimate should better fit the actual value. Nevertheless, it is worth to think about the mechanisms underlying such atmospheric nitrate deposition seasonal invariance in the literature: was it due to its low values? Or maybe to contradictory seasonalities in the precipitation and emissions (or concentration in the air) binomial? April to September precipitation and nitrate deposition in the Bohemian forest were only 15% and 5% higher than in October-March, respectively, which denotes a fairly stable atmospheric nitrogen concentration, with just slightly higher atmospheric nitrate concentrations
in the winter semester that would be mainly counterbalanced by a lower precipitation. Similarly, stable atmospheric HNO3 and total inorganic nitrogen species concentrations were measured in Chinese background sites along the year, with only particulate NO3 slightly higher in autumn-winter than spring-summer. In the light of these observations, it is likely that





atmospheric nitrate concentration in Ergaki, during the snow season, was either similar to or slightly higher than in the snow-free season. In case of invariable atmosphere nitrate concentrations along the year, higher precipitation during the snow-free

season would trigger also a higher nitrate deposition, and our precipitation-dependent estimate (table 2, 4th row) would be our best estimate. In the hypothetical case where the proportion between snow and snow-free season atmospheric nitrate concentrations was the same as in the Chinese site (1.24), and taking into account the higher snow-free season precipitation in Ergaki (1.95), the resulting snow-free nitrate deposition would be about 1.6 times that in the measured snow season and the yearly load would be about $104\pm62$ mg $NO_3$-N·$m^{-2}$·$y^{-1}$. In conclusion, it is reasonable to think that the actual yearly nitrate

deposition was somewhere in-between the time-dependent and the precipitation-dependent estimates. Indeed, in case of a nitrate emissions seasonality as the one registered in the Chinese site in the literature, our precipitation-dependent estimate would be closer to the actual value.

In the case of sulphate, winter atmospheric deposition in Ergaki ($190\pm91$ mg $SO_4$-S·$m^{-2}$), was about four times higher than background values in Canadian Rocky mountains ($\leq 50$ mg $SO_4$-S·$m^{-2}$) (Wasiuta et al., 2015), but 11-12 times lower than in a

Japanese site receiving sulphate from Chinese coal combustion (Ohizumi et al., 2016). In the former case, winter sulphate deposition was 2-5 times larger than in summer and in the latter 3.5-4 times. Unexpectedly high seasonality in the pristine location was due to much higher precipitation in winter but the atmospheric sulphate concentration was relatively constant along the year. On the other hand, sulphate deposition seasonality in the polluted site can be attributed to higher coal burning and emissions in winter. According to these observations, an area with an intermediate winter sulphate deposition like in Ergaki

is likely to have a somewhat higher winter than summer atmospheric sulphate concentrations. In this case, the precipitation-weighted yearly deposition estimate would be an overestimation but we cannot rigorously determine if the actual value would be above or below the time-weighted estimate. At most, we could orientatively assume a linear relationship between yearly sulphate deposition load and its seasonality. Then, a seasonality 11-12 times lower than in the Japanese site form the literature would imply a yearly deposition load of 251 mg $SO_4$-S·$m^{-2}$ in Ergaki. Accordingly, our time-dependent estimate would be our

best estimate but still an overestimate. In conclusion, for the sake of a simpler discussion, we will only consider the time-weighted estimate of yearly sulphate deposition and precipitation-weighted estimates of phosphorus and nitrate depositions. Nevertheless, these estimates must be interpreted cautiously: Whereas nitrate and sulphate deposition estimates might be slightly overestimated, phosphorus would be underestimated.

The selected yearly deposition rate estimates (table 2, 3rd and 4th rows) were compared to global model predictions from the

literature. A global deposition model predicted c. 100 mg $NO_3$-N·$m^{-2}$·$y^{-1}$ and 100-200 mg $SO_4$-S·$m^{-2}$·$y^{-1}$ loads on West Sayan mountains for year 2000, whereas it forecasted ranges from 50-200 mg $NO_3$-N·$m^{-2}$·$y^{-1}$ and 50-200 mg $SO_4$-S·$m^{-2}$·$y^{-1}$ in 2030, according to different scenarios (Lamarque et al., 2013). Therefore, our 2016-17 nitrate deposition estimate roughly fitted the model, whereas sulphate deposition was clearly higher than expected. In the case of phosphorus deposition, our TP estimate was slightly lower than predicted (c. 2 mg TP-P·$m^{-2}$·$y^{-1}$), although uncertainties linked to pollen contribution could make the

actual TP-P deposition match or even surpass the modelled values. On the other hand, the phosphate fraction would be clearly higher than expected (0.1-0.5 mg $PO_4$-P·$m^{-2}$·$y^{-1}$, Mahowald et al. 2008).





In conclusion, the atmospheric nitrate deposition in Ergaki mountain ridge is at the very low range and is between 5 to 20 times lower than in polluted areas of the world. Nevertheless, it is clearly above the background deposition of 0-50 mg $NO_3$-N·$m^{-2}$·$y^{-1}$, as it used to be the case of most Siberia in 1850 or the Antarctica and unpolluted parts of the oceans in 2000

(Lamarque et al., 2013). As for total phosphorus deposition, the uncertainty linked to non measured spring-summer biogenic and wildfire contributions, makes it hard to position the studied site within a world ranking. Our estimate excluding these important biogenic and wildfire contributions (1.71 mg TP-P· $m^{-2}$·$y^{-2}$) and our primitive guess including them (1.972 mg TP-P· $m^{-2}$·$y^{-2}$) would be lower than any terrestrial measurement, according to a worldwide review (≥3 mg TP-P· $m^{-2}$·$y^{-2}$) (Tipping et al., 2014). Nevertheless, it is also possible that pollen and wildfires accounted for a larger contribution and that the present

study site exceeded the latter value. In any case, atmospheric phosphorus deposition in Ergaki would be above the background values corresponding to the poles and the oceans (≤1-2 mg TP-P· $m^{-2}$·$y^{-2}$ and ≤0.5 mg $PO_4$-P· $m^{-2}$·$y^{-2}$), excluding the Atlantic strip downwind of the Sahara (Mahowald et al., 2008). Finally, our yearly sulphate deposition estimate should be cautiously considered, as it could be overestimated due to expectably lower deposition during summer. In any case, it would positively exceed the background values of 0-50 and 50-100 mg $SO_4$·$m^{-2}$·$y^{-1}$ typical in the polar areas and southern hemisphere oceans,

respectively (Lamarque et al., 2013).

### 3.4 SCPs deposition rate

The calculated SCPs deposition rate in Ergaki Natural Park (312±174 ×$10^3$ SCPs·$m^{-2}$·$y^{-1}$) was high above the background rates recorded in Baikal middle basin (57×$10^3$ SCPs·$m^{-2}$·$y^{-1}$), Svalbard islands (13×$10^3$ SCPs·$m^{-2}$·$y^{-1}$) and Nevada Rocky mountains (1.3±0.8 ×$10^3$ SCPs·$m^{-2}$·$y^{-1}$) (Reinemann et al., 2014; Rose et al., 1998). Indeed, it is also far below the records in more polluted

areas such as lake Paione Superiore in western Alps (40900×$10^3$ SCPs·$m^{-2}$·$y^{-1}$) or a set of north African lakes (1098-23694 ×$10^3$ SCPs·$m^{-2}$·$y^{-1}$), where production of electricity by thermal means has increased in the last years (Rose et al., 1999b, 2003). In comparison to a couple of lakes sampled in 1992 in the Khamar-Daban mountains (Southern Siberia) (262 and 780 ×$10^3$ SCPs·$m^{-2}$·$y^{-1}$), the SCP deposition rate in Ergaki was more similar to the lake that was relatively farther from Irkutsk pollution source (Rose et al., 1998). Our data also falls in the lower range of Tatra mountains (225-5240×$10^3$ SCPs·$m^{-2}$·$y^{-1}$) and the

Pyrenees (229-630 ×$10^3$ SCPs·$m^{-2}$·$y^{-1}$) in the mid-1990s (Rose et al., 1998, 1999a; Šporka et al., 2002). An interesting and paradoxical case to compare with is lake Grånästjärn in 1980, with a very similar SCP deposition (300 ×$10^3$ SCPs·$m^{-2}$·$y^{-1}$) but sulphate and nitrate deposition rates 2.6 and 1.8 times higher than in Ergaki (Bergström et al., 2005; Wik and Renberg, 1996). At a first glance, it could seem that our SCP, sulphate and nitrate data didn't match. Nevertheless, at least sulphate depositions differing up to c. 40% have been observed at a particulate low SCP deposition rate (Rose and Monteith, 2005). Additionally,

sulphate measurements in Sweden might include a higher percentage of marine sulphate than in the heart of Eurasia. Finally, apart from this single Swedish lake where the proportion of nitrate to SCPs differs so much from ours, both SCP and nitrate deposition measured in this study are generally comparable to the low range of values in the literature.





### 3.5 Spatial distribution and origin of atmospheric depositions

The spatial distribution of deposited chemical species showed two different patterns. On the one hand, phosphorus forms and
SCPs showed even distribution between sampling sites. On the other hand, $NO_3$, TN and $SO_4$ depositions were significantly
higher on Tsirkovoe and Oiskoe than on Tushkan (fig. 4). Even distribution of phosphorus deposition suggests a common
source of atmospheric phosphorus for all Ergaki sites. Under very low atmospheric phosphorus deposition, like in Ergaki
mountain ridge, biogenic and combustion origins are more important than mineral (Mahowald et al., 2008). This is supported
by several evidences in our case. Firstly, even if the predominant western air mass retrotrajectories partly traverse Kazakhstan
steppe a percentage of days, they hardly ever cross central Asian deserts (fig. 5 a and b). Moreover, direct contact between air
mass retrotrajectories and the Earth crust occurs more often in the taiga ecoregion (yellow dots, fig. 5 a and b). Finally, the
percentage of TP which is soluble in our snow samples (79%) is comparable to European aerosols (50-100%), but much higher
than Saharan dust (8-25%) (Mahowald et al., 2008), so it suggests that phosphorus aerosols in Ergaki were not of desert origin.
Thus, Ergaki mountain lakes differ from Baikal lake, which can be influenced by dust originating at the Gobi desert (Jambers
and Van Grieken, 1997).







**Figure 4. Half year deposition of some airborne chemical species along with snow at 3 different sites in Ergaki mountain ridge. Bars are average values and whiskers represent standard deviation. Sites with different letters on error bars belong to different groups defined by post-hoc Tukey HSD analysis (one-way ANOVA p-values<0.01).**







**Figure 5. Daily three-day long air mass retrotrajectories flowing onto lake Oiskoe from 1st October 2016 to 5th April 2017 (a). Only retrotrajectories of air masses causing precipitation onto Oiskoe at different zooms (b, c and e) and to Tsirkovoe (d) and Tushkan (f). Yellow dots represent hourly records where the air mass retrotrajectories contacted the Earth crust. Water bodies are in black. Grey lines are either political borders or the road. Sources: Map tiles by Stamen Design, under CC BY 3.0. Data by OpenStreetMap,**
**under ODbL; Airmass retrotrajectories by http://www.ready.noaa.gov.**

Nitrate and sulphate are tracers of fossil fuel or biomass combustion (Mahowald et al., 2008). Their spatial correlation (lower in Tushkan than in the other two sites, fig. 4) suggests that alternative nitrate origins like chemical fertilizers or secondary transformations like nitrification/denitrification by microbes in the snowpack might be unimportant. In the same vein, nitrate and sulphate had a relatively high, although not significant, Pearson correlation coefficient (0.63) in the seven analysed snow
core sections. The reason for lower nitrate and sulphate values in Tushkan is still unclear. This site is a forest glade located 300 m altitude lower than the other two. We argue that this altitude difference is too limited to trigger any differences in nitrate deposition and, in any case, the higher the site, the lower the expected deposition (Dambrine et al., 2018). Nonetheless, shorter distance between Tsirkovoe and Oiskoe and their wind regimes might be more decisive, even if the former is located on the north face and the latter on the south. The wind speeds there were 3.4 m·s$^{-1}$ and 3.3 m·s$^{-1}$, respectively, whereas in Tushkan,
farther from the ridge, it was 3.0 m·s$^{-1}$. Despite wind rose circular correlation between Tsirkovoe and Oiskoe was not significant, their local air mass retrotrajectories were quite more similar than to Tushkan (fig. 5 d, e and f). Perhaps, the two more northern sites might be also more exposed to the dominant snow-forming westerlies that flow between latitudes 52° and 53° and above (fig. 5 c). As for SCPs, which are also originated by combustion, the nitrate and sulphate spatial distribution is not followed. We speculate that the particulate character of SCPs might impose different atmospheric transport properties, so
even if SCPs are good tracers of air pollution, slight mismatches between SCPs and chemical pollutants might occur (Rose and Ruppel, 2015; Wik and Renberg, 1996). Anyway, a trend to higher values in the northernmost sites is also observed (fig. 4).

Finally, local wind speed differences between sites might be a tracer of air mass origin. Generally, faster winds would be capable to deliver chemical species from longer distances. The average retrotrajectory length had a weak, non significant, but
still positive Pearson correlation with nitrate (0.47) and negative with phosphate (-0.48), TP (-0.25) and TN (-0.20). This suggests that nitrate deposited on Ergaki mountains might originate at farther distances than phosphorus and particulate nitrogen. To sum up, we hypothesise that northern cities might contribute more to the nitrate deposition than southern ones. These combustion-produced chemicals would be uploaded to the northernmost half of the dominant westerlies conveyor belt flowing directly onto Ergaki mountains, rather than on the southernmost half of the westerlies flow, which mainly traverse
Kazakhstan and turn northwards to Ergaki mountain ridge just before reaching the city of Kyzyl (fig. 5 c). These urban sources might not only include those at the local scale (Abakan, Minusinsk, Chernogorsk, rather than Kyzyl in the south) but also those at a regional one (central and eastern south Siberian rather than northern Kazakhstan cities).





### 3.6 Nutrient deposition and lake water stoichiometry

The relationship between atmospheric deposition and lake water concentration of nutrients is modulated by an array of
processes occurring in the watershed soils, run-off, rivers, water column, sediments, etc. A non-exhaustive list would include:
nitrification and denitrification in streams and wetlands, phosphorus sorption by sediments and soils, bedrock weathering,
deposition of particulate nutrients on lake sediments and redissolution of nutrients back to the water column, mineralization of
organic nutrients or avoidance of such mineralization via the mycoloop, etc. (Chróst and Siuda, 2002; Grossart et al., 2016;
Schlesinger, 2000). In this section and the following one, we consider all these processes as a black box and focus only on the
eventual relationship between atmospheric nutrient deposition and phytoplankton biomass and/or nutrient chemistry in lake
water, as other studies did before (Bergström and Jansson, 2006; Bergström et al., 2005; Elser et al., 2009).

The molar stoichiometry of precipitation-dependent yearly deposition estimations on Ergaki mountain ridge were 169±105
NO3-N/TP-P, 251±167 NO3-N/PO4-P, and 424±310 TN-N/TP-P (mol/mol). Nevertheless, as already mentioned, the
precipitation dependent nitrate estimation might be overestimated and the phosphorus ones are definitely underestimated. If
we took the abovementioned speculative guesses based on our autumn-winter measurements and its relationship to spring-
summer measurements found in the literature, the molar ratios would be 128±79 NO3-N/TP-P and 190±126 NO3-N/PO4-P
(mol/mol). Of course, any discussion based on these estimates is preliminary and needs to be contrasted with future year-long
measurements of atmospheric nutrient deposition, including snow-free season too.  Nevertheless, in both speculative cases,
the NO3-N/TP-P deposition molar ratio in Ergaki would belong to the higher quartile, as compared to a set of alpine regions
of the world (Brahney et al., 2015) (fig. 6 a). Briefly, it would be due to the extraordinary low phosphorus deposition as
compared to nitrogen, which was also relatively low but not that much. As no other DIN form but nitrate was detected in our
samples NO3-N/TP-P ratio can be compared to DIN-N/TP-P ratios in the literature. The closest nutrient deposition
stoichiometries to that of Ergaki (precipitation-dependent estimates) were at Sant Nicolau valley in the Pyrenees (170.5 DIN-
N/TP-P, molar) and the Tatra mountains (165.8) (Brahney et al., 2015; Kopáček et al., 2000, 2011b; Ventura et al., 2000). The
Tatra and Pyrenees lakes used to be P-limited at the turn of the millennium, when they had these atmospheric nutrient
deposition stoichiometries. However, a more comprehensive meta-analysis in the Pyrenees, covering the 1998-2010 period,
showed a lower average DIN-N/TP-P (c. 116) and a trend to shift into potential nitrogen limitation (Camarero and Catalan,
2012). Also the Tatra mountains value in Brahney's review should be considered cautiously because it is based on DIN wet
deposition during 1990-94 and average TP wet deposition during 1998-2009. Average values for the latter time interval in
both, Tatra mountains and Bohemian forest, also had a lower DIN-N/TP-P ratio (98). On the other hand, the alternative 128±79
NO3-N/TP-P (mol/mol) deposition estimate for Ergaki would relocate our lake district closer to the other dots in the graph
(fig. 6 a), with southern Sweden lake district as the closest one (125.6). In any case, Ergaki atmospheric nutrient deposition
stoichiometry is by far larger than that of northern Sweden (20.1 molar DIN-N/TP-P), the paradigm of pristine areas with very
low anthropogenic atmospheric nitrogen deposition and naturally nitrogen limited lakes.



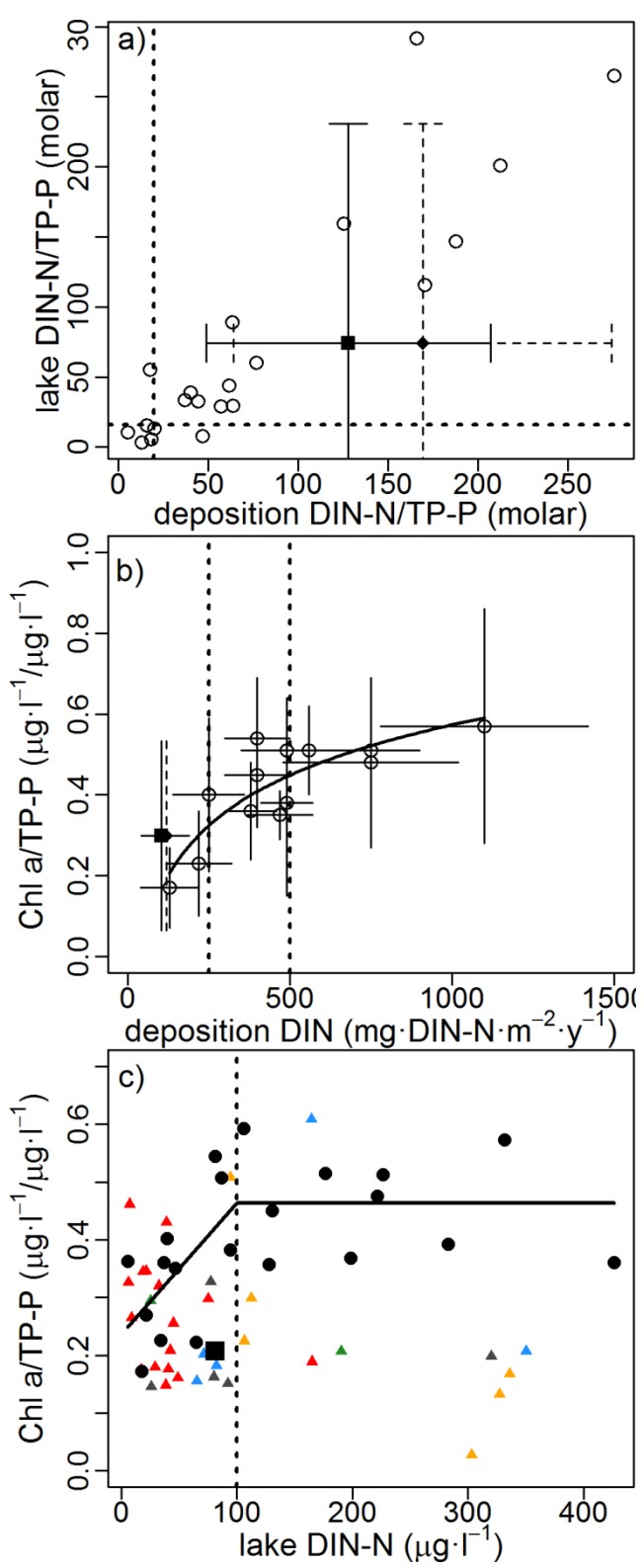





FIGURE 6. The relationship between atmospheric nutrient deposition, nutrient bioavailability and phytoplankton biomass in Ergaki lakes (black square) in the context of different world data sets. a: Atmospheric deposition versus lake DIN-N/TP-P molar ratios of several alpine regions of the world, as reproduced from (Brahney et al., 2015), including Ergaki means and standard deviations: precipitation-dependent estimates (rhombus and dashed lines) and literature corrected estimate (square and solid lines). Vertical dashed line represents the referential north Sweden value (20.1). Horizontal dashed line represents the Redfield ratio (16). b: Yearly atmospheric DIN-N deposition versus Chl a/TP-P ratio of 13 Swedish regions, as reproduced from (Bergström and Jansson, 2006; Bergström et al., 2005), including Ergaki means and standard deviations (as before). Dashed lines represent approximate limits of N, N-P and P limitation areas, according to the authors (250 and 500 mg DIN-N$\cdot$m$^{-2}\cdot$y$^{-1}$). c: Lake DIN-N concentration versus Chl a/TP-P ratio of different lake districts in the world (black circles) and its segmented regression fit as reproduced from (Camarero and Catalan, 2012), including Ergaki medians (black square) and particular observations from the following lakes (triangles): Tsirkovoe (yellow), Oiskoe (red), Raduzhnoe (blue), Karovoe (grey) and Svetloe (green). Dashed line represents limit between N and P limitation conditions, according to the authors.

In the same vein, the average molar stoichiometry of lake water samples was widely above the Redfield ratio (50±128 DIN-N/TP-P, fig. 6 a, table S2), which suggests lake phytoplankton could rather be P limited. Nonetheless, it is important to pay attention to the high dispersion of our data. The ratios in the 2012 survey were outliers one order of magnitude higher than in the other years, stretching the mean upwards (table 3). Indeed, there was a trend to turn from potential P (early June and August 2011-12) to N or N-P colimitation (early September 2015 and late August 2017) in all the studied lakes but in Tsirkovoe, if we compare the available nutrient ratio to the Redfield ratio (16:1, table 3). Besides to the Redfield criterion, a previous study determined that DIN-N/TP-P molar ratios of 3.3, 4.9 and 7.5 would correspond to 75%, 50% and 25% probabilities of chlorophyll increase under a N enrichment experiment (Bergström, 2010). According to this more restrictive criterion, only lake Oiskoe would approach the 75% probability of having a positive increase of chlorophyll under N enrichment in the last two surveys (2015, 2017). In conclusion, the potential nutrient limitation regime seems to be variable in the studied lakes with a possible trend towards N limitation. Unfortunately, it is not possible to make more robust conclusions on this topic without a systematic longer term monitoring.

| Year | month | Lake | DIN-N/TP-P (mol/mol) | TN/TP (mol/mol) | Limiting nutrient |
|------|-------|------|----------------------|-----------------|-------------------|
| **2011** | early June and August | Oiskoe | 19.1 | | P |
| | | Svetloe | 18.7 | | |
| | | Raduzhnoe | 71.3 | | |
| | | Karovoe | 28.1 | | |
| | | mean | 34.3 | | |
| **2012** | early June and August | Oiskoe | 422.9 | | P |
| | | Svetloe | 194.4 | | |



| | | | | | |
|---|---|---|---|---|---|
| | | Raduzhnoe | 176.9 | | |
| | | Karovoe | 38.1 | | |
| | | mean | 208.1 | | |
| **2015** | early September | Oiskoe | 3.7 | 11.6 | N |
| | | Raduzhnoe | 10.5 | 30.9 | N |
| | | Karovoe | 23.6 | 82.8 | P |
| | | mean | 12.6 | 41.8 | N |
| **2017** | late August | Oiskoe | 3.8 | 35.8 | N |
| | | Karovoe | 16.3 | 39.1 | N-P |
| | | Tsirkovoe | 61.9 | 103.7 | P |
| | | mean | 27.3 | 59.5 | P |

**Table 3. Lake water N:P ratios along the studied years and limiting nutrient according to nutrient availability (DIN-N/TP-P) and the Redfield ratio.**

The molar stoichiometry of volume weighted mean concentrations, i.e. the concentrations that would be measured if we had sampled the whole snow core at once, were 169±76 NO3-N/TP-P, 251±134 NO3-N/PO4-P, and 486±357 TN-N/TP-P (mol/mol). To pick up two cases from the literature, the nutrient ratios in the Bothnian Bay (northern Sweden) snowpack were

164±121 NO3-N/TP-P and 326±179 TN-N/TP-P (mol/mol), and on lake Redon (Catalan Pyrenees), 111±106 NO3-N/TP-P, 198±107 DIN-N/TP-P , 118±116 NO3-N/SRP-P and 216±104 TN-N/TP-P (mol/mol) (Catalan, 1989; Rahm et al., 1995). Therefore, NO3-N/TP-P or DIN-N/TP-P in the snowpacks were similar but TN/TP was higher in Ergaki.

### 3.7 Atmospheric input and lake phytoplankton biomass limitation

Atmospheric nitrogen deposition rate was lower in West Sayan mountains than in the most pristine areas in Sweden during the

period 1995-2001, and two orders of magnitude lower than the most impacted Swedish region (Bergström et al., 2013, 2005; Bergström and Jansson, 2006) (fig. 6 b). The Swedish atmospheric deposition gradient was used to establish the new paradigm by which the natural state of many unproductive lakes ($\leq$25 µg TP-P·l$^{-1}$) would be nitrogen limited, when atmospheric deposition is below c. 250 mg DIN-N·m$^{-2}$·y$^{-1}$. According to these authors, unproductive N-limited lakes would shift into N-P colimitation regime under atmospheric depositions of 250-500 mg DIN-N·m$^{-2}$·y$^{-1}$, and into P limitation above 500. Therefore,

West Sayan mountain lakes (79-119 mg DIN-N·m$^{-2}$·y$^{-1}$) would clearly belong to the potential atmospherically induced N limitation domain, where slight increases in atmospheric nitrogen deposition trigger larger phytoplankton biomass shifts.

According to Bergström and colleagues, the Chla/TP-P ratio of N-limited lakes would increase with atmospheric N deposition up to the threshold where they would become P-limited. Above this threshold, extra nitrogen inputs would not change the Chla/TP-P ratio, and eventual phosphorus inputs would also trigger chlorophyll increases, so no change in the ratio would be

observed. Later on, Camarero and Catalan reckoned this threshold as ~100 µg DIN-N·l$^{-1}$ in lake water (Camarero and Catalan,

**Biogeosciences** Open Access
Discussions
EGU

2012) (fig. 6 c). Despite some of our measurements fell above the threshold, the median of 81 µg DIN-N·l$^{-1}$ confirms N limitation in most of the sampled lakes (table S2). The non normal distribution of lake DIN in our data made median a better estimate of the central value. Thus, the vicinity of West Sayan lakes to the threshold suggests that an eventual moderate increase in DIN could lead to a 2-3 fold phytoplankton biomass increase and a state change into P-limitation regime and consequent

phytoplankton community shifts.

The future dynamics of atmospheric nutrients deposition and phytoplankton limitation regime in West Sayan mountain lakes is uncertain. On the one hand, the observed high snow sulphate concentrations and the wind analysis, made us suggest coal combustion in cities of central and eastern south Siberia to be the main winter atmospheric nitrate source. Additionally, predicted winter precipitation in this region under the RCP4.5 scenario (IPCC Representative Concentration Pathway scenario

assuming 4.5 W·m$^{-2}$ radiative forcing by 2100) would increase 10-30% in 2016-2035 and up to 20-30% in 2081-2100 (IPCC, 2013). Therefore, it is likely that atmospheric nitrogen deposition increased even in a scenario where actual emissions did not change. This would probably push lake DIN above the mentioned threshold and would trigger a phytoplankton shift. On the other hand, wild forest fire events in south central Siberia have multiplied and intensified during the last decades and are expected to follow this trend in the 21$^{st}$ century as well (Brazhnik et al., 2017; Malevsky-Malevich et al., 2008). Their effect

on atmospheric nutrient dynamics will be complex. Apart from modifying the sources of phosphorus rich biogenic aerosol particles, wildfires themselves used to be considered as a nitrogen volatilization pulse that left phosphorus on the burnt land (Hungate et al., 2003; Raison, 1979). Nevertheless, a recent study unveiled their relevance as a source of atmospheric phosphorus too (Wang et al., 2015). To sum up, uncertainties on the magnitude and timing of future fossil fuel combustion, precipitation and wildfire regimes make it difficult to predict the status of West Sayan mountain lakes, but it is credible that

they were pushed into a higher trophic state and phosphorus limitation regime, with its consequent phytoplankton community shift.

Nevertheless, such phytoplankton changes might likely depend on temperature as it was the case in the pristine Swedish north (Bergström et al., 2013). The atmospheric nitrogen deposition in the Swedish region declined to < 100 mg DIN-N·m$^{-2}$·y$^{-1}$ in 2011, virtually as deposition in West Sayan lakes, but summer lake water temperatures were roughly 5-18 °C there and 5-14

°C here in West Sayan lakes. They found clear phytoplankton responses to experimental NH$_4$NO$_3$ additions only in warm enough and N-limited lakes. We speculate that relatively lower temperatures in West Sayan lakes might have impeded the Chl a/TP-P ratio to increase so far (fig. 6 c), but the expected warming in the region (IPCC, 2013) could unlock the phytoplankton biomass increase even under a steady-state atmospheric nitrogen deposition scenario.

## 4. Conclusions

The Ergaki Natural Park in the West Sayan mountains was reckoned as an above-background but low atmospheric nutrient deposition area. Our atmospheric total phosphorus and nitrate deposition estimates reasonably fitted those predicted by global deposition models, whereas sulphate and phosphate proved to be higher than expected. While nitrogen values were comparable

to the lowest records in other mountain areas of the world, phosphorus deposition was likely at the very lowest range ever measured on terrestrial ecosystems before. Nevertheless, any conclusions regarding yearly atmospheric phosphorus deposition

loads from this study should be contrasted with further year-long measurements in West Sayan mountains, including the presumably important biogenic and wildfire contributions during spring and summer seasons. In conclusion, the atmospheric reactive nitrogen deposition load was similar to that in northern Sweden, so low enough to potentially induce nitrogen limitation in lake phytoplankton. However, according to our best estimate of phosphorus deposition, the stoichiometry of atmospheric nutrient deposition could be highly determined by the extreme low phosphorus deposition. As a result, the

deposited DIN-N/TP-P ratio would be at the upper quartile of alpine regions worldwide and similar to southern Sweden, where phytoplankton in low productive lakes was generally phosphorus limited. As a synthesis of these contradictory theses, the observed lake nutrient stoichiometry and nutrient limitation regime resulted to be variable, with an apparent trend over the years and/or along the seasonal succession from phosphorus to nitrogen limitation.

This frontier status between phosphorus and nitrogen limitation and the low atmospheric deposition rates make West Sayan

mountain lakes very sensitive to any shift in nutrient deposition. For example, the expected precipitation and wildfires increase could push these lakes to the phosphorus limitation realm, increased phytoplankton biomass and community change, in case that fossil fuel combustion was not limited. Nevertheless, such changes could be temperature limited in a way that lakes were on the eve of a temperature unlocking of primary production and a consequent change of state in the ecosystems. Therefore both, phytoplankton biomass and species composition in these lakes might serve as helpful early warnings of global and

regional environmental change.

Lastly, further studies on the effect of atmospheric nutrient deposition on Siberian mountain lakes, namely including the whole year long deposition, are strongly recommended. The pre-industrial lake phytoplankton nitrogen limitation paradigm was based on a particular region of the world: first in Sweden and then Norway, Colorado, etc. (Bergström et al., 2013, 2005; Bergström and Jansson, 2006; Elser et al., 2009). Nevertheless, increasing atmospheric phosphorus deposition from Saharan origin on the

Pyrenees was able to reduce lake DIN, despite increasign atmospheric nitrogen deposition (Camarero and Catalan, 2012). This, along with the extremely low snow-season atmospheric phosphorus deposition measured in this study opens the possibility to hypothesise variations in the general paradigm, there where atmospheric phosphorus depositions were particularly high or low. Thus, in case that the whole year long atmospheric nitrogen to phosphorus deposition ratio kept high enough due to low phosphorus deposition, phytoplankton in Siberian mountain lakes at pre-industrial times could have been either phosphorus

limited or, simply, insignificantly affected by atmospheric inputs.

**Data availability**

The data sets presented in this study are available as supplementary tables S1 and S2 at the repository of the library of the Siberian Federal University: http://elib.sfu-kras.ru/handle/2311/135098**.**



## Team list

Diaz-de-Quijano, Daniel; Ageev, Aleksander Vladimirovich; Ivanova, Elena Anatolevna; Anishchenko, Olesia Valerevna

## Author contribution

DD and EAI designed the snow sampling, DD and AVA performed it and DD and OVA analysed snow samples. All the authors participated in sampling and sample analyses of lake water. DD prepared the manuscript with contributions from all co-authors.

## Competing interests

The authors declare that they have no conflict of interest.

## Acknowledgements

The authors gratefully acknowledge the NOAA Air Resources Laboratory (ARL) for the provision of the HYSPLIT transport and dispersion model and READY website (http://www.ready.noaa.gov) used in this publication. We also would like to thank the direction of the Ergaki Natural Park for their support and willingness to collaborate.

## Financial Support

The reported study was funded by the Russian Foundation for Basic Research (RFBR), project number 20-04-00960 and the program 5-100 of the Ministry of Science and Higher Education of the Russian Federation.

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
