# Peer review of "Winter atmospheric nutrients and pollutants deposition on West"

_Biogeosciences, 2020_

## Referee Comment (RC1) · Anonymous Referee #2 · 12 Oct 2020

General comments

The manuscript topic falls within the scope of BG. It presents interesting data from an unexplored region. I think it is a valuable contribution on a relevant scientific topic i.e. pollutant/nutrient deposition in remote areas and the possible effects on the ecology of mountain lakes. The results are reported in a clear way but some sections could be shortened and presented more concisely. Some more information on lake features and lake chemical data could be provided (see specific comments).

Specific comments

Lines 47-48: There is no mention here and in the manuscript of the modelled deposition estimates made by EMEP (Co-operative programme for moni-
toring and evaluation of the long-range transmission of air pollutants in Europe;
https://www.emep.int/mscw/index.html): I would suggest the authors to consider these
estimates and possibly compare them with the measured deposition deriving from their
snowpack analyses. I think that s could be an added value to the paper.

Line 54: "warmed": do the author mean subject to global warming?

Some more information could be provided on the lake sites e.g. in Tab. S2, such as
lake surface area and depth, land cover. This information could help in understand the
differences in nutrient levels among the lakes. Deposition is indeed a relevant but not
the unique driver of nutrients lake water.

Line 122: please specify sampling depths

Lines 122-124: the authors used data from a previous lake surveys: Were sampling
ad analytical methods comparable with the present study? For instance, the sampling
period was slightly different in the two surveys (June-Aug in 2011-2012, Aug-Sept in
2015-2017): could this affect the differences in water chemistry between the two sur-
veys (see comment below about Table S2)

Lines 236-237: less than 50% of TN is in the form of $NO_3$. Because $NH_4$ and $NO_2$
are negligible, the remaining part is organic N, Is there an hypothesis for such a high
amount of the organic part? The comparison with deposition at other remote sites
(lines 216-234) could consider also the relevance of inorganic vs organic N (if these
information are available for the mentioned sites e.g. Pyrenees, Alps, Sierra Nevada).

Tab. 2: It should be briefly mentioned in the table caption that "$\sim$ time" and "$\sim$ precip-
itation" referred to different approaches for estimated deposition, and then referred to
the text for the explanation.

Lines 238-243: $SO_4$ values are indeed quite high. The authors stated that these values
are possibly overestimated because referred only to the winter period: why deposition

should be "expectably lower during summer" (line 353)? Do the authors totally exclude long-range transport form large sources, which could explain this high SO4 deposition?

Paragraph 3.3 I would suggest reorganising this paragraph and shorten it. The comparison of the deposition estimates of the present study (Tab.2) with other studies or with global deposition models could be eventually summarised in a table in the SM.

Lines 284-295: Personally, I think this paragraph does not add any useful information on the estimate of P deposition and could be skipped. As the authors said, the use of pollen is an inaccurate method for the estimate: type and coverage by vegetation, meteorological features, and other factors should be considered. Furthermore, other sources than pollen could contribute to P deposition.

Lines 300-305: I agree that a seasonality in NO3 deposition could be scarcely evident at remote sites with very low deposition rates. However, precipitation amount is probably more important at these sites in shaping the seasonal pattern of deposition.

Lines 360-361: The cited site in the Alps was an example of a remote site affected by long-range transport of air pollutants from the lowlands. Furthermore, the SCP values referred to periods of markedly high pollutant deposition (1980s-ealy 1990s). This holds for many sites, at least in Europe, where deposition of air pollutants, especially SO4, decreased significantly in the last 3 decades. I would suggest considering this temporal discrepancy when making the comparison with other sites. Conclusions: this paragraph ca be shortened too, also because the content is partly already provided in the discussion. Conclusions can be maybe provided in the form of a few concise statements summarising the main outcomes of the study and the future research needs.

Tab. 1: I would speak about "local pollution sources" more than "local perturbations"

Table S2:

- SO4 is lacking. It could be interesting to see the SO4 level in lake water, considering the quite high atmospheric input of SO4 estimated form snowpack analysis.

- Further, there are quite sharp differences in some variables (e.g. NO3, TP) between the 2011-2012 and the present survey e.g. NO3 in Oiskoe and Raduzhnoe was markedly higher in the first survey. On the opposite, TP seem to be significantly higher in the most recent survey. Could this be due to the different sampling procedure (composite vs grab surface sampling) or to the slightly different period of the year?

- TP values are quite high, especially in Oiskoe in 2015, pointing to a mesotrophic status of the lake: is there any hypothesis for that? Deposition is discussed in the manuscript as a P input, but these values lead to hypothesised other inputs (catchment sources)

---

## Referee Comment (RC2) · Anonymous Referee #1 · 13 Oct 2020

Review of Diaz-de-Quijano et al. for Biogeosciences

In the paper "Winter atmospheric nutrients and pollutants deposition on West Sayan mountain lakes (Siberia)" Diaz-de-Quijano and coauthors determined the nutrients (nitrates, total phosphorus, and sulphate) and the pollutant spheroidal carbonaceous particles (SCPs) in snowpacks of a remote, poorly known mountains in Siberia (West Sayan) only during the snow period. Then, they estimated using two approaches (time-weighted and precipitation-weighted) the annual deposition of nutrients and SCPs in the region. The ultimate goal is to know if this region is out of relevant nitrogen precipitation but submitted to climatic warming. Finally, they assessed the relevance of these

inputs of N and P on lake nutrients and chlorophyll-a. I found the paper too extended in some parts and very speculative in other ones. I have several comments/concerns that I details below.

Main concerns: - I think the calculations to obtain the annual atmospheric deposition are too speculative and a focus in the real numbers could have been more productive, accurate and direct. - The consequences of the atmospheric deposition of nutrients and pollutants for the lakes are poorly evaluated.

Minor concerns Abstract- line 20, the authors stated that the lakes have "a trend toward nitrogen limitation", despite the N:P molar ratio of atmospheric deposition is very high . This sentence seems to me counterintuitive.

Introduction- Line 33, the word "paradigmatically" seems to me inappropriate Line 35, similar comments the word "paradigmatically" seems to me inappropriate Line 43, this sentence seems to be not proper in scientific, technical writing

Methods- Lines 63 to 88. The description of the study site is too long. Figure 2 seems to me more appropriate to be Figure 1. The first thing to explain should be the location of the study site and then the meteorological characteristics (not climatic). My suggestion is to change the order of Figure 2 and Figure 1. Table 1- I was unable to see the site Tushkan in the map (current Figure 2). Include also the numbers in table 1

Results and discussion- Line 174, meaning acronym SWE Line 196, delete "had" Line 217, 191+/- 35 please being consistent with the data in Table 2. The comparison here seems to me very forced mostly considering the standard deviations of the values. Line 238 please delete "a little bit" that is too colloquial Table 2, please insert units in columns and rows Line 250, this affirmation is only true in humid climates. Dry deposition could be more relevant for instance in the Mediterranean climate. Line 254, please delete "a bit" that is too colloquial Line 258, please delete "as a rule of thumb" Lines 257 to 295, these paragraphs are too speculative. Is the P-linked to pollen available? Line 347, please delete "our primitive guess" Line 368, please delete "At a first

glance" Lines 489-526, I have some concerns about phytoplankton limitation based on data of atmospheric deposition. Lake, or better phytoplankton, limitation should take in account lake stoichiometry and corroborate phytoplankton limitation using bioassays. Taking atmospheric deposition, as a surrogate of lake limitation needs to be better augmented. It is too speculative and needs an experimental approach or more lake data. TP encompasses available and not available P.

––––––––––––––––––––––––––––––––

---

## Author Comment (AC1) · 24 Nov 2020

General comments
The manuscript topic falls within the scope of BG. It presents interesting data from an
unexplored region. I think it is a valuable contribution on a relevant scientific topic i.e.
pollutant/nutrient deposition in remote areas and the possible effects on the ecology of
mountain lakes. The results are reported in a clear way but some sections could be
shortened and presented more concisely. Some more information on lake features and
lake chemical data could be provided (see specific comments).

Specific comments

**REFEREE #2 COMMENT 1:**

Lines 47-48: There is no mention here and in the manuscript of the mod-
elled deposition estimates made by EMEP (Co-operative programme for monitoring
and evaluation of the long-range transmission of air pollutants in Europe;
https://www.emep.int/mscw/index.html): I would suggest the authors to consider these
estimates and possibly compare them with the measured deposition deriving from their
snowpack analyses. I think that s could be an added value to the paper.

**AUTHORS ANSWER 1:**

We are preparing a revised manuscript. We added the references there (lines 47-48) and included the
EMEP deposition model results for 2017 in the discussion (line 337, former line numbers). The EMEP
deposition estimates are within the ranges of Lamarque and colleagues (2013), so they don't modify our
main conclusions. The EMEP deposition model is more accurate in time (for year 2017, when we
sampled the snowpack) but, unfortunately, less accurate in space because our sampling site is located 200
Km east from the EMEP deposition map boundaries.

**REFEREE #2 COMMENT 2:**

Line 54: "warmed": do the author mean subject to global warming?

**AUTHORS ANSWER 2:**

Yes, we changed that sentence (line 54). It is now: "According to published global models (IPCC, 2013;
Lamarque et al., 2013), the West Sayan mountains, in south central Siberia, correspond to a low
atmospheric nitrogen deposition area with a cold but increasingly warming climate in the last decades"

**REFEREE #2 COMMENT 3:**

Some more information could be provided on the lake sites e.g. in Tab. S2, such as
lake surface area and depth, land cover. This information could help in understand the
differences in nutrient levels among the lakes. Deposition is indeed a relevant but not
the unique driver of nutrients lake water.

**AUTHORS ANSWER 3:**

We fully agree with this comment: the role of the watershed is crucial and might explain differences in
water chemistry between lakes. We suggest to include a new supplementary table (Table S1 in the new
manuscript version) with the fields: lake name, coordinates, altitude, maximum depth, Secchi disk,

subsurface chlorophyl a concentration, lake area, watershed area, area of the lake/area of the watershed, watershed land use/land cover area %. Official whole Russia or Krasnoyarsk Territory vegetation cover and soil maps are not enough detailed for the purposes of our study (see the attached Atlas of Specially Protected Natural Territories of the Siberian Circle of the Russian Geographic Society, 2012, pp. 248-249). Therefore, detailed watershed land cover/land use maps have been manually defined for each lake. Polygons have been defined using QGIS 3.14.16-Pi on the basis of Google Satellite and Open Street Map XYZ tiles. Lake, whole watershed and watershed cover areas have been calculated using ellipsoidal project.

Oiskoe and Svetloe are relatively large forest lakes (0.57 and 0.37 Km$^2$, 21 and 24 m maximum depth) with low water transparency (4 and 8 m Secchi disk, respectively). Tsirkovoe, Raduzhnoe and Karovoe are located at an alpine landscape and are smaller and shallower (0.02, 0.03, and 0.08 Km$^2$, respectively; 15, 4 and 7m deep, respectively). Raduzhnoe and Karovoe lake beds were visible. Secchi disk was not tested at Tsirkovoe. Karovoe and Svetloe lakes represent a 7% of the watershed area, Tsirkovoe and Oiskoe, a 5%, and Raduzhnoe is only a 1.4% of its watershed area.

As for vegetation cover, Oiskoe and Svetloe watersheds have a 25 and 28% forest cover whereas the other lakes have less than 10% forest covers. These two watersheds are quite similar in terms of land cover: they have quite equilibrated percentages of forests, shrubs, meadows and scree. Oiskoe is also the watershed with higher peatland cover (6%), followed by Svetloe (3%) and Radushnoe (1.6%). Karovoe and Raduzhnoe watersheds are dominated by scree (73% and 52%, respectively) and meadows (14% and 24%, respectively), whereas Tsirkovoe watershed is dominated by shrubs (56%) and scree (37%).

The abovementioned information will be included in the study site description. Nevertheless, our aim was not to compare differences between watersheds or lakes but to study a representative group of lakes that informed about regional processes, as far as possible.

Comparison of snapshots of the mentioned atlas and the self-made map for Oiskoe basin:

[Figure]

**REFEREE #2 COMMENT 4:**

Line 122: please specify sampling depths

**AUTHORS ANSWER 4:**

The sampling depths were not homogeneous, as it was reported in table S2. We will add an explicit mention to it and cite table S2 at this point of the text to avoid any misunderstanding.

**REFEREE #2 COMMENT 5:**

Lines 122-124: the authors used data from a previous lake surveys: Were sampling ad analytical methods comparable with the present study? For instance, the sampling period was slightly different in the two surveys (June-Aug in 2011-2012, Aug-Sept in 2015-2017): could this affect the differences in water chemistry between the two surveys (see comment below about Table S2)

**AUTHORS ANSWER 5:**

Yes, sampling periods are different and that might explain a particular percentage of the differences in water chemistry and chlorophyll across years because ecological succession was at different stages in different years samplings. Of course, the non-systematic sampling is an important minus of our lake chemistry data set. For that reason, we clearly state that limitation of our data set here in the methods section (lines 122-125), as well as in the discussion (lines 471-472 and 478-479) and in Table S2 (sample column). See discussion and graph below, on this same point.

As for analytical methods, there NO3, NO2 and NH4 were analysed according to the Russian standard method in 2011 and 2012. In this method the water sample is filtered through a paper filter. In 2015 and 2017 the same method was used but using an 0.45 m pore membrane filter. In spite of the fact that porus size was not the same in the case of paper filter, only free ions would react in the analysis, so results are comparable. The important difference in DIN values for 2011-12 and 2015-17 were not due to due to different analytical methods but to a mistake between ionic and element units. The same occurred with TP in 2017. We have amended that mistakes in the new version of the manuscript. Now results look different, sections 3.6, 3.7 and conclusions will be changed. The corrected versions of figure 6 and table 3 will be as follows:

[Figure]

New figure 6

| Year | month | Lake | DIN-N/TP-P (mol/mol) | TN/TP (mol/mol) | Limiting nutrient |
|------|-------|------|----------------------|-----------------|-------------------|
| 2011 | early June and August | Oiskoe | 4.3 | | N |
| | | Svetloe | 4.3 | | N |
| | | Raduzhnoe | 16.2 | | N-P |
| | | Karovoe | 6.5 | | N |
| | | mean | 7.9 | | N |
| 2012 | early June and August | Oiskoe | 109.2 | | P |
| | | Svetloe | 49.5 | | P |
| | | Raduzhnoe | 42.1 | | P |
| | | Karovoe | 29.6 | | P |
| | | mean | 57.6 | | P |
| 2015 | early September | Oiskoe | 3.7 | 11.6 | N |
| | | Raduzhnoe | 10.5 | 30.9 | N |
| | | Tsirkovoe | 23.6 | 82.8 | P |
| | | mean | 12.6 | 41.8 | N |
| 2017 | late August | Oiskoe | 6 | 56.6 | N |
| | | Karovoe | 25.8 | 61.9 | P |
| | | Tsirkovoe | 97.9 | 164.1 | P |
| | | mean | 43.3 | 94.2 | P |

New table 3

Basically, the new conclusions are that the study site is a typical low atmospheric nitrogen deposition area, with lower deposition than the northern Sweden average, as it was described in the seminal paper by Bergström and colleagues where they formulated a new paradigm for phytoplankton growth limitation in oligotrophic lakes (figure 6 b). The studied West Sayan district is safely located in the nitrogen limited realm (figure 6 c). The idea that atmospheric nutrient deposition is quite unimportant for lake water chemistry and phytoplankton growth in these lakes is confirmed by the fact that DIN-N/TP-P ratios of atmospheric deposition and lake water clearly differ (figure 6 a). In conclusion, according to our data, both nitrogen and phosphorus limiting conditions occurred in the studied West Sayan mountain lakes (new table 3 and new figure 6 a) but the region as a whole would be predominantly located at the nitrogen limiting realm (new figure 6 a and c) and constitutes an excellent site to study the effects of global warming with a relative independence of atmospheric nitrogen deposition.

**REFEREE #2 COMMENT 6:**

Lines 236-237: less than 50% of TN is in the form of NO3. Because NH4 and NO2 are negligible, the remaining part is organic N, Is there an hypothesis for such a high amount of the organic part? The comparison with deposition at other remote sites (lines 216-234) could consider also the relevance of inorganic vs organic N (if these information are available for the mentioned sites e.g. Pyrenees, Alps, Sierra Nevada).

**AUTHORS ANSWER 6:**

We didn't pay special attention to this fact, as organic nitrogen is not directly usable as a nutrient by lake phytoplankton. Nevertheless, it might be interesting to mention that, because organic nitrogen is especially important in the studied Ergaki mountain snowpack. We suggest to add a paragraph like this:

"The organic nitrogen in Ergaki snowpack represented a $56\pm19\%$ of the total nitrogen, which is a high but reasonable value as compared to the literature. The relative share of organic to total nitrogen in the snowpacks of the Pyrenees, Alps and Sierra Nevada (USA) mountains, as well as on the Baltic sea were $10\pm9\%$, $41\pm13\%$, $49\pm17\%$ and $21\pm12\%$, respectively (Catalan, 1989; Clement et al., 2012; Pearson et al.,

2015; Rahm et al., 1995). Organic nitrogen has been reported elsewhere to be higher in snowpack records than in wet deposition because dry deposition of organic nitrogen is integrated in the snowpack and because microbial uptake and assimilation of inorganic nitrogen might occur in the snowpack (Clement et al., 2012; Pearson et al., 2015). In the case of microbial conversion from inorganic to organic nitrogen, it could be hypothesised that deeper and older snow layers should have higher organic nitrogen shares. Such a pattern was only observed at Tushkan (ANOVA, p-value= 0.0178) whereas no significant differences were found in the percentatge of organic nitrogen between upper and lower snow layers at Oiskoe and Tsirkovoe sites. We, therefore, hypothesise that a different combination of phenomena might be responsible for orgnaic nitrogen dominance at different sites."

[Figure]

**Column pairs with "a" and "b" letters are significantly different (one-way ANOVA, p-v<0.05; n=3 except in Tsirkovoe, where upper layer n=4 and lower layer n=2).**

**REFEREE #2 COMMENT 7:**

Tab. 2: It should be briefly mentioned in the table caption that "_ time" and "_ precipitation" referred to different approaches for estimated deposition, and then referred to the text for the explanation.

**AUTHORS ANSWER 7:**

OK, we will add these sentences to the table caption:
"Yearly deposition rates were estimated on the basis of measured winter depositions and either assuming a constant deposition rate (time weighted estimate, row 3) or a precipitation-dependent deposition rate (precipitation-weighted estimate, row 4). See section 3.3 for further discussion."

**REFEREE #2 COMMENT 8:**

Lines 238-243: SO4 values are indeed quite high. The authors stated that these values are possibly overestimated because referred only to the winter period: why deposition

should be "expectably lower during summer" (line 353)? Do the authors totally exclude long-range transport form large sources, which could explain this high SO4 deposition?

**AUTHORS ANSWER 8:**

We suggest that the high SO4 values in the snowpack could be due to combustion of coal, which is commonly used for domestic and central heating in villages and cities at a regional scale. Therefore, we do not exclude long-range transport from large sources. That is discussed in section 3.5.We will change the phrase "Finally, our yearly sulphate deposition estimate should be cautiously considered, as it could be overestimated due to expectably lower deposition during summer" by:

"Finally, our yearly sulphate deposition estimate should be cautiously considered, as it could be overestimated due to regionally widespread coal combustion for heating during winter (see section 3.5)."

**REFEREE #2 COMMENT 9:**

Paragraph 3.3 I would suggest reorganising this paragraph and shorten it. The comparison of the deposition estimates of the present study (Tab.2) with other studies or with global deposition models could be eventually summarised in a table in the SM.

**AUTHORS ANSWER 9:**

OK, that is true. This section is written in an inductive way. It will be reformulated to a deductive structure of the speech, which will probably be more communicative and, hopefully, also shorter. A summarising table with the deposition estimates of the present study and those of global deposition models will be added in the SM if the manuscript is allowed to pass to the following revision step.

**REFEREE #2 COMMENT 10:**

Lines 284-295: Personally, I think this paragraph does not add any useful information on the estimate of P deposition and could be skipped. As the authors said, the use of pollen is an inaccurate method for the estimate: type and coverage by vegetation, meteorological features, and other factors should be considered. Furthermore, other sources than pollen could contribute to P deposition.

**AUTHORS ANSWER 10:**

Well, that is true that this paragraph does not add any especially valuable information but it rather reinforces the idea that the previous TP deposition estimate could be a credible value (or, alternatively, an underestimate). Moreover, it has also been criticized by the other referee. The paragraph was not present in the original manuscript but added on request of a reviewer in a previous submission to another journal. It will be deleted it if the editor allows us to submit a new version of the manuscript.

**REFEREE #2 COMMENT 11:**

Lines 300-305: I agree that a seasonality in NO3 deposition could be scarcely evident at remote sites with very low deposition rates. However, precipitation amount is probably more important at these sites in shaping the seasonal pattern of deposition.

**AUTHORS ANSWER 11:**

Yes, remote sites have a low seasonal variation of atmospheric NO3 concentration, so the seasonal NO3 deposition is basically ruled by precipitation seasonality in these remote and humid environments. We concluded that from the detailed discussion of a couple of study cases with seasonal information on

atmospheric NO3 deposition and precipitation (in Czech Republic) and atmospheric NO3 concentrations (China) (lines 300-317). That is why we chose the precipitation-weighted estimate of yearly NO3 deposition (lines 330-331).

**REFEREE #2 COMMENT 12:**

Lines 360-361: The cited site in the Alps was an example of a remote site affected by long-range transport of air pollutants from the lowlands. Furthermore, the SCP values referred to periods of markedly high pollutant deposition (1980s-ealy 1990s). This holds for many sites, at least in Europe, where deposition of air pollutants, especially SO4, decreased significantly in the last 3 decades. I would suggest considering this temporal discrepancy when making the comparison with other sites. Conclusions: this paragraph ca be shortened too, also because the content is partly already provided in the discussion. Conclusions can be maybe provided in the form of a few concise statements summarising the main outcomes of the study and the future research needs.

**AUTHORS ANSWER 12:**

We added the specification that data from the Alps refers to the more polluted times of 1980s and early 1990s. We also reviewed that all the comparisons with values found in the literature always included the information about the years when they were measured, if distant from the publication date. On the other hand, we suggest not to shorten this paragraph, as it is the only one in the discussion where we compare our calculated SCPs deposition rate to those in the literature.

OK. Conclusions will be shortened.

**REFEREE #2 COMMENT 13:**

Tab. 1: I would speak about "local pollution sources" more than "local perturbations"

**AUTHORS ANSWER 13:**

OK. We changed it.

**REFEREE #2 COMMENT 14:**

Table S2:
- SO4 is lacking. It could be interesting to see the SO4 level in lake water, considering the quite high atmospheric input of SO4 estimated form snowpack analysis.

**AUTHORS ANSWER 14:**

We have added SO4 measurements for lake water in the lake water chemistry table at SM. Values range from 0 to 1900 µg SO4-S/l. In 2015 it was not measured.

**REFEREE #2 COMMENT 15:**

- Further, there are quite sharp differences in some variables (e.g. NO3, TP) between the 2011-2012 and the present survey e.g. NO3 in Oiskoe and Raduzhnoe was markedly higher in the first survey. On the opposite, TP seem to be significantly higher in the most recent survey. Could this be due to the different sampling procedure (composite vs grab surface sampling) or to the slightly different period of the year?

**AUTHORS ANSWER 15:**

Yes, that was true according to the submitted manuscript. As we already mentioned, we found a mistake in our lake water chemistry data. We did two mistakes. Firstly, we took 2011 and 2012 DIN data as N-NO3, N-NO2 and N-NH4 but in fact the ionic forms had been measured (NO3, NO2 and NH4). Secondly, we did also a mistake when converting from total ionic PO4 (after digestion) measured in 2017 into TP. All the units of the original data and the necessary conversion factors have been checked during this revision of the manuscript. After the corrected data, the difference in DIN levels between 2011-12 and 2015-17 is much more moderate than it appeared to be before. As for TP, changes are minor.

In any case, even after the corrected data, the question posed by the reviewer remains completely pertinent. As we previously said, the non-systematic sampling is a minus of our data set and we need to be very cautious when extracting any conclusion from that. Nevertheless, the data set is still informative. We expose our rationale here:

All sources of variability are important: composite vs. surface water sampling (sampling), stage of yearly plankton succession (season), and year variability (year). Oiskoe lake was sampled as composite samples (2011 and 2012) and at different depths (2015 and 2017). The differences between composite or discrete depth sampling can be minimised. The separate values obtained from different depths at 2015 and 2017 expeditions can be averaged to simulate a composite sample analysis and make data comparable across years and plankton succession stages. This cannot be done in the case of Raduzhnoe where samplings were either composite sample (2011) or surface water (2012-15). Nevertheless, Raduzhnoe is a small shallow lake. Its maximum depth at a very particular place in the middle of the lake, amongst rock boulders at the bottom of the lake, is 4m but most of the lake is generally no more than 2 m deep. Therefore, mixing might be important in this lake and, consequently, the differences in chemical composition are likely to be moderate between different depth water layers.

The other two sources of variability (season and year) cannot be disentangled in any way. That is a limitation of our data set. Nevertheless, there are some patterns that can be mentioned (see graph below):

- Both, at Oiskoe and Raduzhnoe lakes, late summer NO3 was lower than early summer NO3, and/or 2015-17 NO3 was lower than 2011-12 NO3.
- Both, at Oiskoe and Raduzhnoe NO3 values were intermediate in 2011 (early and mid summer), higher in 2012 (early and mid summer) and low in 2015 and 2017 (late summer). If we removed 2012 outlier, the trend still would be to decrease in time (months and/or years).
- In the case of TP, there is a high variability in the values within any of the plotted lakes and there is no clear temporal trend (along succession and/or years). It should be noted that the TP values corresponding to 2015 are slightly higher than in the other years. TP in 2015 survey samples was measured differently from all the other cases: instead of directly analyzing TP, they were the sum of dissolved inorganic phosphorus (DIP) and particulate phosphorus (PP). This information was missing in the former manuscript version (!) but it has been included in the next version.

In conclusion, N tends to decrease in time (although it is uncertain if it is a seasonal and/or an interannual trend), but because P values oscillate, the N:P ratio also behaves this way (see corrected table 3 above in this document).

[Figure]

Lake water nitrate and total phosphorus concentrations at different time points of the plankton succession sampled at different years: 2011 (red), 2012 (yellow), 2015 (green) and 2017 (purple).

**REFEREE #2 COMMENT 16:**

- TP values are quite high, especially in Oiskoe in 2015, pointing to a mesotrophic status of the lake: is there any hypothesis for that? Deposition is discussed in the manuscript as a P input, but these values lead to hypothesised other inputs (catchment sources)

**AUTHORS ANSWER 16:**

Yes, as we stated in Table 1, Oiskoe has the most human-modified watershed including an inflow from nearby houses. We add a new phrase underlying this fact also in the discussion, in former line 477. We will add:
"The effects of atmospheric nutrient deposition could be blurred by watershed processes at this lake [Oiskoe]. Forest and peat cover (25% and 6%, respectively) are important in comparison to the other lakes and, more importantly, the lake receives an inflow passing by nearby houses (table 1). Nevertheless, that houses and land covers were present during the whole studied period and the local anthropogenic impact-free Karovoe and Raduzhnoe lakes also showed a decrease in DIN-N/TP-P ratios in time."

---

## Author Comment (AC2) · 24 Nov 2020

Review of Diaz-de-Quijano et al. for Biogeosciences

In the paper "Winter atmospheric nutrients and pollutants deposition on West Sayan mountain lakes (Siberia)" Diaz-de-Quijano and coauthors determined the nutrients (nitrates, total phosphorus, and sulphate) and the pollutant spheroidal carbonaceous particles (SCPs) in snowpacks of a remote, poorly known mountains in Siberia (West Sayan) only during the snow period. Then, they estimated using two approaches (time-weighted and precipitation-weighted) the annual deposition of nutrients and SCPs in the region. The ultimate goal is to know if this region is out of relevant nitrogen precipitation but submitted to climatic warming. Finally, they assessed the relevance of these inputs of N and P on lake nutrients and chlorophyll-a. I found the paper too extended in some parts and very speculative in other ones. I have several comments/concerns that I details below.

**REFEREE #1 COMMENT 1:**

**Main concerns:** - I think the calculations to obtain the annual atmospheric deposition are too speculative and a focus in the real numbers could have been more productive, accurate and direct. - The consequences of the atmospheric deposition of nutrients and pollutants for the lakes are poorly evaluated.

**AUTHORS ANSWER 1:**

We agree that our manuscript combines an empirical (snow period) and a speculative (snow-free period) component regarding atmospheric nutrients and pollutants deposition. Probably it would have been a faster and more simple option just to show snowpack chemistry and stop there. Nevertheless, we think that this speculative exercise is legitimate and valuable for two reasons. First of all, because it is clearly and honestly separated from the empirical measurements. Secondly, because only the estimated yearly load allowed us to compare our study site with other lake districts in the literature, in terms of the relationship between atmospheric nutrient deposition and lake phytoplankton limitation regime. By doing that, we could get to interesting general conclusions.

At this point of the answers to the comments of the reviewers we have to say that during the revision process we found two mistakes that have changed the lake water chemistry dat aset in a way that conclusions have changed, too. Nevertheless, the yearly load estimation remains necessary to identify the location of the phytoplankton of the studied lakes in the nitrogen to phosphorus limitation gradient and its relationship to atmospheric nutrient deposition.

After making the necessary corrections, figure 6 and table 3 will be changed into the following correct versions:

[Figure]

New figure 6

| Year | month | Lake | DIN-N/TP-P (mol/mol) | TN/TP (mol/mol) | Limiting nutrient |
|------|-------|------|-----------------------|------------------|--------------------|
| **2011** | early June and August | Oiskoe | 4.3 | | N |
| | | Svetloe | 4.3 | | N |
| | | Raduzhnoe | 16.2 | | N-P |
| | | Karovoe | 6.5 | | N |
| | | mean | 7.9 | | N |
| **2012** | early June and August | Oiskoe | 109.2 | | P |
| | | Svetloe | 49.5 | | P |
| | | Raduzhnoe | 42.1 | | P |
| | | Karovoe | 29.6 | | P |
| | | mean | 57.6 | | P |
| **2015** | early September | Oiskoe | 3.7 | 11.6 | N |
| | | Raduzhnoe | 10.5 | 30.9 | N |
| | | Tsirkovoe | 23.6 | 82.8 | P |
| | | mean | 12.6 | 41.8 | N |
| **2017** | late August | Oiskoe | 6 | 56.6 | N |
| | | Karovoe | 25.8 | 61.9 | P |
| | | Tsirkovoe | 97.9 | 164.1 | P |
| | | mean | 43.3 | 94.2 | P |

New table 3

Sections 3.6, 3.7 and conclusions will be changed to a shorter text than in the previous version of the manuscript. Basically, the new conclusions are that the study site is a typical low atmospheric nitrogen deposition area, with lower deposition than the northern Sweden average, as it was described in the seminal paper by Bergström and colleagues where they formulated a new paradigm for phytoplankton growth limitation in oligotrophic lakes (figure 6 b). The studied West Sayan district is safely located in the nitrogen limited realm (figure 6 c). The idea that atmospheric nutrient deposition is quite unimportant for lake water chemistry and phytoplankton growth in these lakes is confirmed by the fact that DIN-N/TP-P ratios of atmospheric deposition and lake water clearly differ (figure 6 a). In conclusion, according to our data, both nitrogen and phosphorus limiting conditions occurred in the studied West Sayan mountain lakes (new table 3 and new figure 6 a) but the region as a whole would be predominantly located at the nitrogen limiting realm (new figure 6 a and c) and constitutes an excellent site to study the effects of global warming with a relative independence of atmospheric nitrogen deposition.

**Minor concerns**

**REFEREE #1 COMMENT 2:**

**Abstract-** line 20, the authors stated that the lakes have "a trend toward nitrogen limitation", despite the N:P molar ratio of atmospheric deposition is very high. This sentence seems to me counterintuitive.

**AUTHORS ANSWER 2:**

After correcting the mistakes in the calculations for lake water chemistry none of those trends are visible anymore. We are not going to discuss about temporal trends in the upcoming version of the manuscript. Instead, we will express as we did above: "both nitrogen and phosphorus limiting conditions occurred in the studied West Sayan mountain lakes (new table 3 and new figure 6 a) but the region as a whole would be predominantly located at the nitrogen limiting realm (new figure 6 a and c"

**REFEREE #1 COMMENT 3:**

**Introduction**- Line 33, the word "paradigmatically" seems to me inappropriate Line 35, similar comments the word "paradigmatically" seems to me inappropriate Line 43, this sentence seems to be not proper in scientific, technical writing

**AUTHORS ANSWER 3:**

OK. We changed the sentence in lines 33 and 35 into:

"The effects of atmospheric nitrogen deposition on primary production have been documented in the usually nitrogen-limited terrestrial ecosystems (Bobbink et al., 2010; DeForest et al., 2004; Güsewell, 2004; LeBauer and Treseder, 2008), as well as in commonly phosphorus-limited lakes (Bergström et al., 2005)."

Well, our opinion is that sentence in line 43 is communicative because it shows that ecology as a scientific discipline has long disattended the importance of geographical distribution of ecological processes, but it is not important, so we also changed it into:

"Nevertheless, ecological processes are not homogeneous around the World."

**REFEREE #1 COMMENT 4:**

**Methods-** Lines 63 to 88. The description of the study site is too long. Figure 2 seems to me more appropriate to be Figure 1. The first thing to explain should be the location of the study site and then the meteorological characteristics (not climatic). My suggestion is to change the order of Figure 2 and Figure 1. Table 1- I was unable to see the site Tushkan in the map (current Figure 2). Include also the numbers in table 1

**AUTHORS ANSWER 4:**

Details concerning the vegetal cover at the study site (paragraph 2, lines 78-88) can be deleted, especially if we also remove the confirmation of our yearly TP deposition estimate based on pollen. This part was not in the original article but added on request of a referee in a previous submission to a different journal.

We changed figure 2 to be Figure 1.

We now say meteorological instead of climatic (lines 76 and 90)

Tushkan is visible in the map as letter C, as it is stated in the caption.

We now added an extra column in table 1 to make it easier to identify sampling sites in the map.

**Results and discussion-**

**REFEREE #1 COMMENT 5:** Line 174, meaning acronym SWE

**AUTHORS ANSWER 5:** Snow water equivalent. Done.

**REFEREE #1 COMMENT 6:** Line 196, delete "had"

**AUTHORS ANSWER 6:** Done

**REFEREE #1 COMMENT 7:** Line 217, 191+/- 35 please being consistent with the data in Table 2. The comparison here seems to me very forced mostly considering the standard deviations of the values.

**AUTHORS ANSWER 7:** That is true. We replaced "higher than" for "comparable to"

**REFEREE #1 COMMENT 8:** Line 238 please delete "a little bit" that is too colloquial Table 2, please insert units in columns and rows

**AUTHORS ANSWER 8:** deleted and done

**REFEREE #1 COMMENT 9:** Line 250, this affirmation is only true in humid climates. Dry deposition could be more relevant for instance in the Mediterranean climate.

**AUTHORS ANSWER 9:** OK, we added "in wet climates like that in West Sayan mountains"

**REFEREE #1 COMMENT 10:** Line 254, please delete "a bit" that is too colloquial

**AUTHORS ANSWER 10:** replaced by slightly

**REFEREE #1 COMMENT 11:** Line 258, please delete "as a rule of thumb"

**AUTHORS ANSWER 11:** replaced by "In general terms"

**REFEREE #1 COMMENT 12:** Lines 257 to 295, these paragraphs are too speculative. Is the P-linked to pollen available?

**AUTHORS ANSWER 12:** There are two different questions here. Lines 257-283 use information on seasonal deposition of atmospheric phosphorus in the literature to evaluate which of our two estimates (based on constant deposition in time and based on precipitation) would be more likely to be true. The same kind of rationale is used in the case of nitrate and sulphate in the following paragraphs in the manuscript. As we said in the answer to the main concerns, we think that such an exercise is valuable to contextualize our study site in comparison to other lake districts of the world in terms of atmospheric nitrogen deposition and lake phytoplankton limitation.

Lines 284-295 have also been criticized by the other referee. They were not present in the original manuscript but added on request of a reviewer in a previous submission to another journal. The paragraph will be deleted if the editor allows us to submit a new version of the manuscript.

Yes, of course, the P linked to pollen is available in a span of articles, as we cited them in the paragraph: (Banks and Nighswander, 2000; Bigio and Angert, 2018; Brown and Irving, 1973; Doskey and Ugoagwu, 1992). Note that we used data at the genus level, not species. In any case, we will delete it.

**REFEREE #1 COMMENT 13:** Line 347, please delete "our primitive guess"

**AUTHORS ANSWER 13:** Replaced by "our literature-based estimate"

**REFEREE #1 COMMENT 14:** Line 368, please delete "At a first glance"

**AUTHORS ANSWER 14:** We replaced "At a first glance, it could seem" by "This may lead to think"

**REFEREE #1 COMMENT 15:** Lines 489-526, I have some concerns about phytoplankton limitation based on data of atmospheric deposition. Lake, or better phytoplankton, limitation should take in

account lake stoichiometry and corroborate phytoplankton limitation using bioassays. Taking atmospheric deposition, as a surrogate of lake limitation needs to be better aug- mented. It is too speculative and needs an experimental approach or more lake data. TP encompasses available and not available P.

**AUTHORS ANSWER 15:** We agree that the lake water chemistry (and stoichiometry) data set that we are using is limited in amount of observations but it is also novel for an underrepresented part of the World and, therefore, valuable. We strongly disagree with the idea that phytoplankton growth limitation only could be assessed using a combination of stoichiometry and bioassays. There are, at least, three different approaches to assess phytoplankton limitation in the environment. Each of them is supported by a vast number of published papers and have been used combined and alone. Firstly, there are enrichment experiments and calculation of response ratios that can be run at the whole lake level but also at *in situ* mesocosms, or *in vitro* in the lab. Secondly, there is a bunch of biochemical or molecular indicators of nutrient limitation in phytoplankton including enzyme activities (e.g., phosphatases, peptidases, etc.), nutrient uptake kinetics, pigment ratios, nucleic acid ratios, membrane transporters, NIFTs, etc. Finally, it is also legitimate to use a stoichiometric approach that can include ratios between different dissolved, particulate or dissolved and particulate nutrients. Of course, discrepancies exist between approaches because of empirical reasons but also because the aspects of phytoplankton growth limitation that we can assess using the different approaches are inherently different. Thus, methods assessing phytoplankton limitation at the ecosystem or cellular level can be perfectly contradictory, as particular cells can be nutrient-limited whereas other cells within the same population or other species within the same community might be not limited at all. Therefore, we stand for the legitimacy of the current approach. Moreover, we include information from previous studies that combined stoichiometry and nutrient enrichment experiments in our discussion.

The cause-effect chain presence or absence between atmospheric nutrient deposition, nutrient concentrations in lake water, and phytoplankton growth are assessed in figure 6 and the discussion associated to that figure. In fact, figure 6 b and c (discussed in lines 489-526 in the former manuscript version) consist in adding one dot corresponding to our study site to graphs representing regional or World scale studies published in very good journals, where lake stoichiometry was presented without any complementing bioassay. The use of Chl a/TP-P ratio is not our innovation but a ratio previously used in the cited articles. The fact that TP might include not bioavailable P is not a problem at all because the gist of figure 6 b and c is to show the relationship between atmospheric and lake water inorganic nitrogen and phytoplankton biomass (represented by chlorophyll a). TP is at the denominator just to remove the effects of changing P on the increase or decrease of Chla and assess the effect of N on Chla alone. Moreover, dissolved inorganic PO4 concentrations in lake water can be below the detection limit in a range of unproductive high mountain lakes like the ones included in the present and cited studies because the phosphate turnover is very fast. For that reason, TP is a more reliable measure of the phosphorus state in ultraoligotrophic lakes than dissolved inorganic phosphorus. Besides that, phytoplankton and general microplankton living in P-limited environments have developed strategies to use phosphorus forms others than dissolved inorganic orthophosphate. These arguments have been used by Brahney and colleagues (2015), Camarero & Catalan (2012) and elsewhere. A short justification will be added to the next manuscript version.

Finally, in the case of figure 6 c and the corresponding discussion, atmospheric deposition is not present at all. We use both graphs, with atmospheric N deposition (figure 6 b) and with lake water DIN (figure 6 c) namely to saw the logical chain between the atmosphere, lake and phytoplankton. Therefore, it is not true that atmospheric N was used as a surrogate of lake limitation.

---

## Author Response (AR1)

General comments
The manuscript topic falls within the scope of BG. It presents interesting data from an
unexplored region. I think it is a valuable contribution on a relevant scientific topic i.e.
pollutant/nutrient deposition in remote areas and the possible effects on the ecology of
mountain lakes. The results are reported in a clear way but some sections could be
shortened and presented more concisely. Some more information on lake features and
lake chemical data could be provided (see specific comments).

Specific comments

**REFEREE #2 COMMENT 1:**

Lines 47-48: There is no mention here and in the manuscript of the mod-
elled deposition estimates made by EMEP (Co-operative programme for monitoring
and evaluation of the long-range transmission of air pollutants in Europe;
https://www.emep.int/mscw/index.html): I would suggest the authors to consider these
estimates and possibly compare them with the measured deposition deriving from their
snowpack analyses. I think that s could be an added value to the paper.

**AUTHORS ANSWER 1:**

We added the references there (line 49) and included the EMEP deposition model results for 2017 in the
discussion (lines 348-351, new line numbers). The EMEP deposition estimates are within the ranges of
Lamarque and colleagues (2013), so they don't modify our main conclusions. The EMEP deposition
model is more accurate in time (for year 2017, when we sampled the snowpack) but, unfortunately, less
accurate in space because our sampling site is located 200 Km east from the EMEP deposition map
boundaries.

**REFEREE #2 COMMENT 2:**

Line 54: "warmed": do the author mean subject to global warming?

**AUTHORS ANSWER 2:**

Yes, we changed that sentence (line 54). It is now: "According to published global models (IPCC, 2013;
Lamarque et al., 2013), the West Sayan mountains, in south central Siberia, correspond to a low
atmospheric nitrogen deposition area with a cold but increasingly warming climate in the last decades"

**REFEREE #2 COMMENT 3:**

Some more information could be provided on the lake sites e.g. in Tab. S2, such as
lake surface area and depth, land cover. This information could help in understand the
differences in nutrient levels among the lakes. Deposition is indeed a relevant but not
the unique driver of nutrients lake water.

**AUTHORS ANSWER 3:**

We fully agree with this comment: the role of the watershed is crucial and might explain differences in
water chemistry between lakes. We included a new supplementary table (Table S1 in the new manuscript
version) with the fields: lake name, coordinates, altitude, maximum depth, Secchi disk, subsurface

chlorophyl a concentration, lake area, watershed area, area of the lake/area of the watershed, watershed land use/land cover area %. Official whole Russia or Krasnoyarsk Territory vegetation cover and soil maps are not enough detailed for the purposes of our study (see the attached Atlas of Specially Protected Natural Territories of the Siberian Circle of the Russian Geographic Society, 2012, pp. 248-249, below). Therefore, detailed watershed land cover/land use maps were manually defined for each lake. Polygons were defined using QGIS 3.14.16-Pi on the basis of Google Satellite and Open Street Map XYZ tiles. Lake, whole watershed and watershed cover areas were calculated using ellipsoidal project.

Oiskoe and Svetloe are relatively large forest lakes (0.57 and 0.37 $Km^2$, 21 and 24 m maximum depth) with low water transparency (4 and 8 m Secchi disk, respectively). Tsirkovoe, Raduzhnoe and Karovoe are located at an alpine landscape and are smaller and shallower (0.02, 0.03, and 0.08 $Km^2$, respectively; 15, 4 and 7m deep, respectively). Raduzhnoe and Karovoe lake beds were visible. Secchi disk was not tested at Tsirkovoe. Karovoe and Svetloe lakes represent a 7% of the watershed area, Tsirkovoe and Oiskoe, a 5%, and Raduzhnoe is only a 1.4% of its watershed area.

As for vegetation cover, Oiskoe and Svetloe watersheds have a 25 and 28% forest cover whereas the other lakes have less than 10% forest covers. These two watersheds are quite similar in terms of land cover: they have quite equilibrated percentages of forests, shrubs, meadows and scree. Oiskoe is also the watershed with higher peatland cover (6%), followed by Svetloe (3%) and Radushnoe (1.6%). Karovoe and Raduzhnoe watersheds are dominated by scree (73% and 52%, respectively) and meadows (14% and 24%, respectively), whereas Tsirkovoe watershed is dominated by shrubs (56%) and scree (37%).

The abovementioned information was included in the study site description (lines 117-129) and in the discussion (lines 470-474). That contributed to a more understandable discussion of results. Nevertheless, we did not dig deep in that direction as our aim was not to compare differences between watersheds or lakes but to study a representative group of lakes that informed about regional processes, as far as possible.

Comparison of snapshots of the mentioned atlas and our self-made map for Oiskoe basin:

[Figure]

[Figure]

**REFEREE #2 COMMENT 4:**

Line 122: please specify sampling depths

**AUTHORS ANSWER 4:**

The sampling depths were not homogeneous, as it was reported in table S2. We added an explicit mention to it and cited table S2 at this point of the text (lines 130-131) to avoid any misunderstanding.

**REFEREE #2 COMMENT 5:**

Lines 122-124: the authors used data from a previous lake surveys: Were sampling ad analytical methods comparable with the present study? For instance, the sampling period was slightly different in the two surveys (June-Aug in 2011-2012, Aug-Sept in 2015-2017): could this affect the differences in water chemistry between the two surveys (see comment below about Table S2)

**AUTHORS ANSWER 5:**

Yes, sampling periods are different and that might explain a particular percentage of the differences in water chemistry and chlorophyll across years because ecological succession was at different stages in different years samplings. Of course, the non-systematic sampling is an important minus of our lake chemistry data set. For that reason, we clearly state that limitation of our data set here in the methods section (lines 130-134) and in Table S2 (sample column). See authors answer 15 below for a full discussion the significance of differences in water column sampling strategy at different surveys.

As for analytical methods, we detailed the differences in the different surveys in the new section 2.2 (lines 136-158). Soluble reactive phosphorus, NO3, NO2 and NH4 were analysed according to the Russian standard method in 2011 and 2012. In this method the water sample is filtered through a paper filter. In 2015 and 2017 the same method was used but using an 0.45 m pore membrane filter. In spite of the fact that pore size was not the same in the case of paper filter, only free ions would react in the analysis, so

results are comparable. Additionally, in 2015, nutrients were measured using a Flow Injection Analyser Lachat Quickchem 8500 autoanalyzer Series 2 FIA System (Hach Ltd, Loveland, CO, U.S.), which has different detection limits as compared to the manual assessment of samples. Details are stated in the new manuscript version. Nevertheless, the important difference in DIN values between 2011-12 and 2015-17 in the former manuscript version were not due to due to different analytical methods or sampling stradies and periods but to a mistake between ionic and element units conversion. The same occurred with TP in 2017. We have amended that mistakes in the new version of the manuscript. Now results look different: figure 6 and new table 3, sections 3.6, 3.7 and conclusions have been consequently changed.

Basically, the new conclusions are that the study site is a typical low atmospheric nitrogen deposition area, with lower deposition than the northern Sweden average, as it was described in the seminal paper by Bergström and colleagues where they formulated a new paradigm for phytoplankton growth limitation in oligotrophic lakes (figure 6 b). The studied West Sayan district is safely located in the nitrogen limited realm (figure 6 c). The idea that atmospheric nutrient deposition is quite unimportant for lake water chemistry and phytoplankton growth in these lakes is confirmed by the fact that DIN-N/TP-P ratios of atmospheric deposition and lake water clearly differ (figure 6 a). In conclusion, according to our data, both nitrogen and phosphorus limiting conditions occurred in the studied West Sayan mountain lakes (new table 3 and new figure 6 a). The region constitutes an excellent site to study the effects of global warming with a relative independence of atmospheric nitrogen and phsophorus deposition.

**REFEREE #2 COMMENT 6:**

Lines 236-237: less than 50% of TN is in the form of NO3. Because NH4 and NO2 are negligible, the remaining part is organic N, Is there an hypothesis for such a high amount of the organic part? The comparison with deposition at other remote sites (lines 216-234) could consider also the relevance of inorganic vs organic N (if these information are available for the mentioned sites e.g. Pyrenees, Alps, Sierra Nevada).

**AUTHORS ANSWER 6:**

We didn't pay special attention to this fact, as organic nitrogen is not directly usable as a nutrient by lake phytoplankton. Nevertheless, it might be interesting to mention that, because organic nitrogen is especially important in the studied Ergaki mountain snowpack. We added a new paragraph to focus on this part of our results (lines 267-277). The following graph might help to read the mentioned paragraph but we think that it is not necessary to include it in the article.

[Figure]

Column pairs with "a" and "b" letters are significantly different (one-way ANOVA, p-v<0.05; n=3 except in Tsirkovoe, where upper layer n=4 and lower layer n=2).

**REFEREE #2 COMMENT 7:**

Tab. 2: It should be briefly mentioned in the table caption that "_ time" and "_ precipitation" referred to different approaches for estimated deposition, and then referred to the text for the explanation.

**AUTHORS ANSWER 7:**

OK, we added these sentences to the table caption:
"Yearly deposition rates were estimated on the basis of measured winter depositions and either assuming a constant deposition rate (time weighted estimate, row 3) or a precipitation-dependent deposition rate (precipitation-weighted estimate, row 4). See section 3.3 for further discussion."

**REFEREE #2 COMMENT 8:**

Lines 238-243: SO4 values are indeed quite high. The authors stated that these values are possibly overestimated because referred only to the winter period: why deposition should be "expectably lower during summer" (line 353)? Do the authors totally exclude long-range transport form large sources, which could explain this high SO4 deposition?

**AUTHORS ANSWER 8:**

We suggest that the high SO4 values in the snowpack could be due to combustion of coal, which is commonly used for domestic and central heating in villages and cities at a regional scale. Therefore, we do not exclude long-range transport from large sources. That is discussed in section 3.5.We changed the phrase to: "Finally, our yearly sulphate deposition estimate should be cautiously considered, as it could be overestimated due to expectably lower deposition during summer" by:

"Finally, our yearly sulphate deposition estimate should be cautiously considered, as it could be overestimated due to regionally widespread coal combustion for heating during winter (see section 3.5)." at lines 366-368.

**REFEREE #2 COMMENT 9:**

Paragraph 3.3 I would suggest reorganising this paragraph and shorten it. The comparison of the deposition estimates of the present study (Tab.2) with other studies or with global deposition models could be eventually summarised in a table in the SM.

**AUTHORS ANSWER 9:**

We reformulated this section from inductive to a deductive structure of the speech. The beginning of each paragraph in the current version contains the conclusion of that same paragraph. That shortened the text from 1959 to 1586 words and made it, hopefully, more communicative.
The literature values with which we compared our results in section 3.2 were too fragmentary to be included in a table. The most important modelized values for our discussion in section 3.3 have been included in Table 2 as new 5$^{th}$ and 6$^{th}$ rows to make the comparison more comfortable to the reader.

**REFEREE #2 COMMENT 10:**

Lines 284-295: Personally, I think this paragraph does not add any useful information on the estimate of P deposition and could be skipped. As the authors said, the use of pollen is an inaccurate method for the estimate: type and coverage by vegetation, meteorological features, and other factors should be considered. Furthermore, other sources than pollen could contribute to P deposition.

**AUTHORS ANSWER 10:**

Well, that is true that this paragraph did not add any especially valuable information but it rather reinforced the idea that the previous TP deposition estimate could be a credible value (or, alternatively, an underestimate). Moreover, it was also criticized by the other referee. The paragraph was not present in the original manuscript but added on request of a reviewer in a previous submission to another journal. The paragraph was deleted in the current version of the manuscript.

**REFEREE #2 COMMENT 11:**

Lines 300-305: I agree that a seasonality in NO3 deposition could be scarcely evident at remote sites with very low deposition rates. However, precipitation amount is probably more important at these sites in shaping the seasonal pattern of deposition.

**AUTHORS ANSWER 11:**

Yes, remote sites have a low seasonal variation of atmospheric NO3 concentration, so the seasonal NO3 deposition is basically ruled by precipitation seasonality in these remote and humid environments. We concluded that from the detailed discussion of a couple of study cases with seasonal information on atmospheric NO3 deposition and precipitation (in Czech Republic) and atmospheric NO3 concentrations (China) (lines 311-327). That is why we chose the precipitation-weighted estimate of yearly NO3 deposition (lines 341-342).

**REFEREE #2 COMMENT 12:**

Lines 360-361: The cited site in the Alps was an example of a remote site affected by long-range transport of air pollutants from the lowlands. Furthermore, the SCP values

referred to periods of markedly high pollutant deposition (1980s-ealy 1990s). This
holds for many sites, at least in Europe, where deposition of air pollutants, especially
SO4, decreased significantly in the last 3 decades. I would suggest considering this
temporal discrepancy when making the comparison with other sites. Conclusions: this
paragraph ca be shortened too, also because the content is partly already provided in
the discussion. Conclusions can be maybe provided in the form of a few concise statements
summarising the main outcomes of the study and the future research needs.

**AUTHORS ANSWER 12:**

We added the specification that data from the Alps refers to the more polluted times of 1980s and early
1990s (lines 374-375). We also reviewed that all the comparisons with values found in the literature
always included the information about the years when they were measured, if distant from the publication
date. On the other hand, we suggest not to shorten this paragraph, as it is the only one in the discussion
where we compare our calculated SCPs deposition rate to those in the literature.

Conclusions were rewritten and shortened from 510 to 371 words.

**REFEREE #2 COMMENT 13:**

Tab. 1: I would speak about "local pollution sources" more than "local perturbations"

**AUTHORS ANSWER 13:**

OK. We changed that.

**REFEREE #2 COMMENT 14:**

Table S2:
- SO4 is lacking. It could be interesting to see the SO4 level in lake water, considering
the quite high atmospheric input of SO4 estimated form snowpack analysis.

**AUTHORS ANSWER 14:**

We have added SO4 measurements for lake water in the lake water chemistry table at SM. Values range
from 0 to 1900 µg SO4-S/l. In 2015 it was not measured.

**REFEREE #2 COMMENT 15:**

- Further, there are quite sharp differences in some variables (e.g. NO3, TP) between
the 2011-2012 and the present survey e.g. NO3 in Oiskoe and Raduzhnoe
was markedly higher in the first survey. On the opposite, TP seem to be significantly
higher in the most recent survey. Could this be due to the different sampling procedure
(composite vs grab surface sampling) or to the slightly different period of the year?

**AUTHORS ANSWER 15:**

Yes, that was true according to the previously submitted manuscript version. As we already mentioned,
we found a mistake in our lake water chemistry data. We did two mistakes. Firstly, we took 2011 and
2012 DIN data as N-NO3, N-NO2 and N-NH4 but in fact the ionic forms had been measured (NO3, NO2
and NH4). Secondly, we did also a mistake when converting from total ionic PO4 (after digestion)
measured in 2017 into TP. All the units of the original data (lake water and snow chemistry) and the
necessary conversion factors have been checked during this revision of the manuscript. After the
corrected data, the difference in DIN levels between 2011-12 and 2015-17 is much more moderate than it
appeared to be before. As for TP, changes are minor and the exceptionally high values at Oiskoe in 2015

were interpreted to be connected to human activities around the lake (lines 470-475), as we have already mentioned above.

In any case, even after the corrected data, the question posed by the reviewer remains completely pertinent. As we previously said, the non-systematic sampling is a minus of our data set and we need to be very cautious when extracting any conclusion from that. Nevertheless, the data set is still informative. We expose our rationale here:

All sources of variability are important: composite vs. surface water sampling (sampling), stage of yearly plankton succession (season), and year variability (year). Oiskoe lake was sampled as composite samples (2011 and 2012) and at different depths (2015 and 2017). The differences between composite or discrete depth sampling can be minimised. The separate values obtained from different depths at 2015 and 2017 expeditions can be averaged to simulate a composite sample analysis and make data comparable across years and plankton succession stages. This cannot be done in the case of Raduzhnoe where samplings were either composite sample (2011) or surface water (2012-15). Nevertheless, Raduzhnoe is a small shallow lake. Its maximum depth at a very particular place in the middle of the lake, amongst rock boulders at the bottom of the lake, is 4m but most of the lake is generally no more than 2 m deep. Therefore, mixing might be important in this lake and, consequently, the differences in chemical composition are likely to be moderate between different depth water layers.

The other two sources of variability (season and year) cannot be disentangled in any way. That is a limitation of our data set and we accordingly deleted any discussion about temporal trends in the current version of the manuscript. Therefore, we keep the discussion and graph below in this document for your interest but it is based on a limited data set and it is not crucial for the manuscript itself.

- Both, at Oiskoe and Raduzhnoe lakes, late summer $NO_3$ was lower than early summer $NO_3$, and/or 2015-17 $NO_3$ was lower than 2011-12 $NO_3$.
- Both, at Oiskoe and Raduzhnoe $NO_3$ values were intermediate in 2011 (early and mid summer), higher in 2012 (early and mid summer) and low in 2015 and 2017 (late summer). If we removed 2012 outlier, the trend still would be to decrease in time (months and/or years).
- In the case of TP, there is a high variability in the values within any of the plotted lakes and there is no clear temporal trend (along succession and/or years). It should be noted that the TP values corresponding to 2015 are slightly higher than in the other years. TP in 2015 survey samples was measured differently from all the other cases: instead of directly analyzing TP, they were the sum of dissolved inorganic phosphorus (DIP) and particulate phosphorus (PP). This information was missing in the former manuscript version but it has been included in the current version. In any case, the increase is notably high in Oiske and moderate in Radushnow (where results are comparable to those obtained in early 2011 summer)

In conclusion, N tends to decrease in time (although it is uncertain if it is a seasonal and/or an interannual trend), but because P values oscillate, the N:P ratio also behaves this way (see corrected table 3 in the new manuscript version).

[Figure]

Lake water nitrate and total phosphorus concentrations at different time points of the plankton succession sampled at different years: 2011 (red), 2012 (yellow), 2015 (green) and 2017 (purple).

**REFEREE #2 COMMENT 16:**

- TP values are quite high, especially in Oiskoe in 2015, pointing to a mesotrophic status of the lake: is there any hypothesis for that? Deposition is discussed in the manuscript as a P input, but these values lead to hypothesised other inputs (catchment sources)

**AUTHORS ANSWER 16:**

Yes, there are 3 things to be considered concerning this comment: (1) In the current manuscript version we noticed, after correcting for conversion factor errors, that lake water chemistry was majorly uncoupled from atmospheric deposition. (2) the calculation of TP as the sum of SRP and PP in 2015 mught partly account for the high TP values that year, but (3) TP in 2015 was unprecedentedly high only in the case of Oiskoe lake (for example in Raduzhnoe they were comparable to the early 2011 summer value). Therefore, we conclude that the high Oiskoe 2015 TP values might be due to human activities in the watershed, including an inflow from nearby houses, which are unique to this lake. That was stated in Table 1, and in the phrases that we addee to the discussion (lines 470-475):
"Nevertheless, it is very likely that this case was due to watershed-level processes. Forest and peat cover (25% and 6%, respectively) in Oiskoe watershed are more important than in other lakes watersheds and, more importantly, the lake receives an inflow passing by nearby houses (table 1 and S1). A local phosphorus input from nearby houses could have occurred at Oiskoe in 2015 pointing to mesotrophic

status of the lake. The high TP values could be also partly due to having calculated TP as the sum of SRP and PP instead of measuring it directly in 2015 but only lake Oiskoe recorded unprecedentedly high values, which might be due to the mentioned human activities in its watershed."
* * *
In the paper "Winter atmospheric nutrients and pollutants deposition on West Sayan mountain lakes (Siberia)" Diaz-de-Quijano and coauthors determined the nutrients (nitrates, total phosphorus, and sulphate) and the pollutant spheroidal carbonaceous particles (SCPs) in snowpacks of a remote, poorly known mountains in Siberia (West Sayan) only during the snow period. Then, they estimated using two approaches (time-weighted and precipitation-weighted) the annual deposition of nutrients and SCPs in the region. The ultimate goal is to know if this region is out of relevant nitrogen precipitation but submitted to climatic warming. Finally, they assessed the relevance of these inputs of N and P on lake nutrients and chlorophyll-a. I found the paper too extended in some parts and very speculative in other ones. I have several comments/concerns that I details below.

**REFEREE #1 COMMENT 1:**

**Main concerns:** - I think the calculations to obtain the annual atmospheric deposition are too speculative and a focus in the real numbers could have been more productive, accurate and direct. - The consequences of the atmospheric deposition of nutrients and pollutants for the lakes are poorly evaluated.

**AUTHORS ANSWER 1:**

We agree that our manuscript combines an empirical (snow period) and a speculative (snow-free period) component regarding atmospheric nutrients and pollutants deposition. Probably it would have been a faster and more simple option just to show snowpack chemistry and stop there. Nevertheless, we think that this speculative exercise is legitimate and valuable for two reasons. First of all, because it is clearly and honestly separated from the empirical measurements. Secondly, because only the estimated yearly load allowed us to compare our study site with other lake

districts in the literature, in terms of the relationship between atmospheric nutrient deposition and lake phytoplankton limitation regime. By doing that, we could: (1) understand the very weak effect of atmospheric nutrient deposition on south-central Siberian mountain lakes, and (2) formulate an interesting hypothesis of a Siberian exception to the current paradigm of limitation of lake phytoplankton growth that needs to be assessed in the future.

During the revision process we found two mistakes that have changed the lake water chemistry data set in a way that conclusions have changed, too. Nevertheless, the yearly load estimation remains necessary to identify the location of the phytoplankton of the studied lakes in the nitrogen to phosphorus limitation gradient and its relationship to atmospheric nutrient deposition.

After making the necessary corrections, figure 6 and new table 3, sections 3.6, 3.7 and conclusions have been consequently changed. Basically, the new conclusions are that the study site is a typical low atmospheric nitrogen deposition area, with lower deposition than the northern Sweden average, as it was described in the seminal paper by Bergström and colleagues where they formulated a new paradigm for phytoplankton growth limitation in oligotrophic lakes (figure 6 b). The studied West Sayan district is safely located in the *potential* nitrogen limitation realm (figure 6 c). Nevertheless, according to our data, both nitrogen and phosphorus limiting conditions occurred in the studied West Sayan mountain lakes (new table 3 and new figure 6 a). That was due to the fact that atmospheric phosphorus depositions were not just as nitrogen deposition but extremely low. As a consequence, minute and unbalanced atmospheric nutrient deposition was easily intercepted and modified by watershed processes. Thus, the atmospheric nutrient deposition resulted quite unimportant for lake water chemistry and phytoplankton growth in these lakes as it was confirmed by the fact that DIN-N/TP-P ratios of atmospheric deposition and lake water clearly differed (figure 6 a).

In conclusion, the region constitutes an excellent site to study the effects of global warming with a relative independence of atmospheric nitrogen and phosphorus deposition. Finally, a regime of alternating N and P limitation events could have been the natural pre-industrial conditions for lake phytoplankton in wide regions with extremely low atmospheric P deposition like Siberia.

The (lack of) link between atmospheric nutrient deposition, lake water nutrient stoichiometry, and phytoplankton biomass are logically woven in the mentioned figure, table and associated discussion sections. The aim of our study was to describe the effect of atmospheric nutrient deposition, nutrient load and ratios on phytoplankton growth, so other atmospheric depositions (organic nitrogen, SCPs and sulphate) are also discussed but are considered secondary for the manuscript aims.

**Minor concerns**

**REFEREE #1 COMMENT 2:**

**Abstract-** line 20, the authors stated that the lakes have "a trend toward nitrogen limitation", despite the N:P molar ratio of atmospheric deposition is very high. This sentence seems to me counterintuitive.

**AUTHORS ANSWER 2:**

After correcting the mistakes in the calculations for lake water chemistry none of those trends are visible anymore. This phrase has been deleted. The third paragraph of the abstract has been changed according to the abovementioned changes in data and manuscript figures tables and

sections. We deleted any discussion relative to temporal trends because our lake water data set was not suitable for that.

**REFEREE #1 COMMENT 3:**

**Introduction**- Line 33, the word "paradigmatically" seems to me inappropriate Line 35, similar comments the word "paradigmatically" seems to me inappropriate Line 43, this sentence seems to be not proper in scientific, technical writing

**AUTHORS ANSWER 3:**

OK. We changed the sentence in lines 33 and 35 into:

"The effects of atmospheric nitrogen deposition on primary production have been documented in the usually nitrogen-limited terrestrial ecosystems (Bobbink et al., 2010; DeForest et al., 2004; Güsewell, 2004; LeBauer and Treseder, 2008), as well as in commonly phosphorus-limited lakes (Bergström et al., 2005)." (lines 34-36)

We changed the sentence in former line 43 into:

"Nevertheless, ecological processes are not homogeneous around the World." (line 44)

**REFEREE #1 COMMENT 4:**

**Methods-** Lines 63 to 88. The description of the study site is too long. Figure 2 seems to me more appropriate to be Figure 1. The first thing to explain should be the location of the study site and then the meteorological characteristics (not climatic). My suggestion is to change the order of Figure 2 and Figure 1. Table 1- I was unable to see the site Tushkan in the map (current Figure 2). Include also the numbers in table 1

**AUTHORS ANSWER 4:**

Details concerning the vegetal cover at the study site have been deleted.

We changed figure 2 to be Figure 1.

We now say meteorological instead of climatic (line 83)

Tushkan is visible in the map as letter C, as it is stated in the caption.

We now added an extra column in table 1 to make it easier to identify sampling sites in the map.

**Results and discussion-**

**REFEREE #1 COMMENT 5:** Line 174, meaning acronym SWE

**AUTHORS ANSWER 5:** Snow water equivalent. Done.

**REFEREE #1 COMMENT 6:** Line 196, delete "had"

**AUTHORS ANSWER 6:** Done

**REFEREE #1 COMMENT 7:** Line 217, 191+/- 35 please being consistent with the data in Table 2. The comparison here seems to me very forced mostly considering the standard deviations of the values.

**AUTHORS ANSWER 7:** That is true. We replaced "higher than" for "comparable to"

**REFEREE #1 COMMENT 8:** Line 238 please delete "a little bit" that is too colloquial Table 2, please insert units in columns and rows

**AUTHORS ANSWER 8:** deleted and done

**REFEREE #1 COMMENT 9:** Line 250, this affirmation is only true in humid climates. Dry deposition could be more relevant for instance in the Mediterranean climate.

**AUTHORS ANSWER 9:** OK, we added "in wet climates like that in West Sayan mountains"

**REFEREE #1 COMMENT 10:** Line 254, please delete "a bit" that is too colloquial

**AUTHORS ANSWER 10:** replaced by slightly

**REFEREE #1 COMMENT 11:** Line 258, please delete "as a rule of thumb"

**AUTHORS ANSWER 11:** replaced by "In general terms"

**REFEREE #1 COMMENT 12:** Lines 257 to 295, these paragraphs are too speculative. Is the P-linked to pollen available?

**AUTHORS ANSWER 12:** There are two different questions here. Lines 257-283 in the former manuscript version used information on seasonal deposition of atmospheric phosphorus in the literature to evaluate which of our two estimates (based on constant deposition in time and based on precipitation) would be more likely to be true. The same kind of rationale was used in the case of nitrate and sulphate in the following paragraphs of the manuscript. As we said in the answer to the main concerns, we think that such an exercise is valuable to contextualize our study site in comparison to other lake districts of the world in terms of atmospheric nitrogen deposition and lake phytoplankton limitation.

Former lines 284-295 have also been criticized by the other referee. They were not present in the original manuscript but added on request of a reviewer in a previous submission to another journal. Therefore, the paragraph has been deleted in the current version of the manuscript.

In any case, yes, of course, the P linked to pollen was available in a span of articles, as we used to cite in the now deleted paragraph: (Banks and Nighswander, 2000; Bigio and Angert, 2018; Brown and Irving, 1973; Doskey and Ugoagwu, 1992). Note that we used to use data at the genus level, not species. In any case, it all has been deleted, now.

**REFEREE #1 COMMENT 13:** Line 347, please delete "our primitive guess"

**AUTHORS ANSWER 13:** Replaced by "our literature-based estimate"

**REFEREE #1 COMMENT 14:** Line 368, please delete "At a first glance"

**AUTHORS ANSWER 14:** We replaced "At a first glance, it could seem" by "This may lead to think"

**REFEREE #1 COMMENT 15:** Lines 489-526, I have some concerns about phytoplankton limitation based on data of atmospheric deposition. Lake, or better phytoplankton, limitation should take in account lake stoichiometry and corroborate phytoplankton limitation using bioassays. Taking atmospheric deposition, as a surrogate of lake limitation needs to be better aug- mented. It is too speculative and needs an experimental approach or more lake data. TP encompasses available and not available P.

**AUTHORS ANSWER 15:** We agree that the lake water chemistry (and stoichiometry) data set that we are using is limited in number of observations but it is also novel for an underrepresented part of the World and, therefore, valuable.

We strongly disagree with the idea that phytoplankton growth limitation only could be assessed using a combination of stoichiometry and bioassays. There are, at least, three different approaches to assess phytoplankton limitation in the environment. Each of them is supported by a vast number of published papers and have been used combined and alone. Firstly, there are enrichment experiments and calculation of response ratios that can be run at the whole lake level but also at *in situ* mesocosms, or *in vitro* in the lab. Secondly, there is a bunch of biochemical or molecular indicators of nutrient limitation in phytoplankton including enzyme activities (e.g., phosphatases, peptidases, etc.), nutrient uptake kinetics, pigment ratios, nucleic acid ratios, membrane transporters, NIFTs, etc. Finally, it is also legitimate to use a stoichiometric approach that can include ratios between different dissolved, particulate or dissolved and particulate nutrients. Of course, discrepancies exist between approaches because of empirical reasons but also because the aspects of phytoplankton growth limitation that we can assess using the different approaches are inherently different. Thus, methods assessing phytoplankton limitation at the ecosystem or cellular level can be perfectly contradictory, as particular cells can be nutrient-limited whereas other cells within the same population or other species within the same community might be not limited at all. Therefore, we stand for the legitimacy of the current approach. Moreover, we include information from previous studies that combined stoichiometry and nutrient enrichment experiments in our discussion.

In the current version of the manuscript, the cause-effect chain between atmospheric nutrient deposition, nutrient concentrations in lake water, and phytoplankton growth was broken at the atmosphere to lake water chemical composition (figure 6 a). Therefore, the situation has changed concerning the stated concerns.

Nevertheless, we would like to explain why was that approach legitimate. Atmospheric deposition was not used in any case as a surrogate of lake phytoplankton growth limitation. Opposite to that, the logical chain between atmospheric nutrient deposition, nutrient concentrations in lake water, and phytoplankton growth was (and is) assessed in figure 6 and the discussion associated to that figure (sections 3.6 and 3.7). In fact, figure 6 b and c consist in adding one dot corresponding to our study site to graphs representing regional or World scale studies published in top rated journals, where lake stoichiometry was presented alone without any complementary bioassays. The use of Chl a/TP-P ratio is not our innovation but a ratio previously used in the cited articles. The fact that TP might include not bioavailable P is not a problem at all because the gist of figure 6 b and c is to show the relationship between atmospheric and lake water inorganic nitrogen and phytoplankton biomass (represented by chlorophyll a). TP is at the denominator just to remove the effects of changing P on the increase or decrease of Chla and assess the effect of N on Chla alone. Moreover, dissolved inorganic PO4 concentrations in lake water can be below the detection limit in a range of unproductive high mountain lakes like the ones included in the present and cited studies because the phosphate turnover is very fast. For that reason, TP is a more reliable measure of the phosphorus state in ultraoligotrophic lakes than dissolved inorganic phosphorus. Besides that, phytoplankton and general microplankton living in P-limited environments have developed strategies to use phosphorus forms others than dissolved inorganic orthophosphate. These arguments have been used by Brahney and colleagues (2015), Camarero & Catalan (2012) and elsewhere and are expressed in the current version of the manuscript (lines 445-449). In short, TP

is basically bioavailable in mountain lakes where phytoplankton has been described to use phosphatase, phosphonate hydrolases, fagotrophy, bacterivory and osmotrophy.

Finally, in the case of figure 6 c and the corresponding discussion, atmospheric deposition is not present at all. We use both graphs, with atmospheric N deposition (figure 6 b) and with lake water DIN (figure 6 c) namely to saw the logical chain between the atmosphere, lake and phytoplankton. Therefore, it is not true that atmospheric N was used as a surrogate of lake phytoplankton growth limitation.

In conclusion, because our lake water chemistry and chlorophyl a data set is modest, we agree that we cannot draw categoric conclusions from them --and we do not-- but they are enough to shed some light into the darkness of an understudied region and promote future research on that.